# Polymerase theta repairs persistent G1-induced DNA breaks in S-phase during class switch recombination

Timea Marton[1], Jinglong Wang[2], Amaury Vaysse [1,3], Wei Yu [1,5], Pierre-Henri Commere[4], Quentin Holleville[1], Tristan Espie-Caullet [1], Richard Frock [2] & Ludovic Deriano [1] ✉

Non-homologous end joining (NHEJ) is the primary pathway for repairing G1 phase-induced DNA double-strand breaks (DSBs) during immunoglobulin heavy chain (*Igh*) class switch recombination (CSR) in B lymphocytes. In B cells lacking NHEJ (XRCC4) or DSB end protection (SHLD1), end joining during CSR proceeds through an alternative end-joining pathway. Polymerase theta (Pol θ) is widely regarded as a mediator of this pathway, essential for repairing replication-associated DSBs during mitosis when homologous recombination is unavailable. In this study, we examined CSR in primary B cells lacking XRCC4, SHLD1, and/or Pol θ, revealing two repair pathways: Pol θ-independent productive switching and Pol θ-dependent unproductive switching characterized by end resection, inversion and microhomology. Furthermore, we show that Pol θ-mediated repair under NHEJ-deficiency coincides with G1-to-S phase transition and occurs independently of RHINO and PLK1. Thus, in the absence of NHEJ, Pol θ repairs persistent G1-phase DSBs during S-phase rather than mitosis.

Polymerase theta (Pol θ), encoded by the *Polq* gene in mice, is a specialized error-prone DNA polymerase playing a significant role in microhomology (MH)-mediated end-joining, an alternative end-joining (alt-EJ) pathway[1–3]. Pol θ is composed of a distinct N-terminal helicase domain capable of displacing replication protein A (RPA)[4], a C-terminal polymerase domain able to generate short stretches of nascent DNA[5] and templated insertions[6,7] and a disordered central region in part necessary for regulation via phosphorylation[8]. Remarkably conserved across species except fungi, Pol θ plays a critical role in DNA repair through its ability to align 3′ overhangs containing MH[9] and repair DNA ends, often resulting in small insertions or deletions that contribute to genomic instability. Over the past years, Pol θ has emerged as a significant therapeutic target in cancers deficient in homologous recombination (HR), such as those with BRCA1/2 mutations[10]. These cancer cells exhibit "Pol θ addiction", relying on Pol θ-mediated end-joining (TMEJ) for survival under conditions of replicative stress. Recognizing its therapeutic potential, several pharmaceutical companies have developed Pol θ inhibitors, some of which are now advancing through clinical trials, offering a promising avenue for treating HR-deficient cancers.

TMEJ is predominantly recognized as a DNA repair mechanism utilized by cells that cannot resolve DNA lesions through HR due to detrimental mutations in HR-associated genes[11,12]. However, recent findings have highlighted the critical role of TMEJ during mitosis (M)[8,13]. In repair-proficient cells, HR and non-homologous end joining (NHEJ) are typically repressed during the M phase of the cell cycle, implying that persistent DNA breaks remain unresolved during this cell phase[14]. Such unsolved DNA damage poses a significant threat to

[1]Institut Pasteur, Université Paris Cité, INSERM U1223, Équipe Labellisée Ligue Contre Le Cancer, Genome Integrity, Immunity and Cancer Unit, Paris, France. [2]Department of Radiation Oncology, School of Medicine, Stanford University, Stanford, CA, USA. [3]Institut Pasteur, Université Paris Cité, Bioinformatics and Biostatistics Hub, Paris, France. [4]Flow Cytometry Platform, Institut Pasteur, Université Paris Cité, Paris, France. [5]Present address: BioBank, The First Affiliated Hospital of Xi'an Jiaotong University, Xi'an, Xi'an, P.R. China. ✉e-mail: lderiano@pasteur.fr

genomic stability during cell division. Unlike HR or NHEJ, TMEJ is active during this phase and enables cells to repair residual double-strand breaks (DSBs) before completing cell division. This mitotic-repair process is orchestrated by the multifunctional activity of Polo-like Kinase 1 (PLK1), whose phosphorylation events activate Pol θ and recruit it to mitotic DSBs[8]. Recruitment relies on the RHINO/TOPBP1 complex[8,13]. HR-deficient cells, such as BRCA2-null cells, face a unique vulnerability. These cells are incapable of repairing S/G2- induced DSBs during mitosis if either Pol θ or RHINO, a 911-complex interactor, is absent[13]. This deficiency underpins the synthetic lethality observed between BRCA mutations and Pol θ, as HR-deficient cells rely heavily on TMEJ to resolve replication-associated stress and maintain survival in the absence of functional HR pathways.

Independent of the cell cycle phase, lingering unrepaired DNA breaks are highly toxic, potentially leading to genetic loss, chromosomal translocations, and the formation of oncogenic lesions[15]. G1-induced breaks depend heavily on NHEJ due to limited DNA-end processing, the absence of resection, and the lack of templates for HR. Analogously to HR- deficient cells, in NHEJ-deficient cells Pol θ also plays a crucial compensatory role in repairing DNA damages of various nature[7,11,16]. Disruption of both NHEJ (e.g., via KU, XRCC4, LIG4 knockouts, or DNA-PK inhibition) and Pol θ results in synthetic lethality[7,11,16,17]. However, the precise timing and mechanisms underlying this TMEJ dependency remain unclear[1].

B lymphocytes are a model for studying NHEJ due to their reliance on programmed G1- induced DNA breaks during V(D)J recombination and class switch recombination (CSR). V(D)J recombination, critical for antigen receptor diversity, enables the rearrangement of variable (V), diversity (D), and joining (J) gene segments[18,19] via the channeling of RAG (RAG1/RAG2)-induced DNA breaks quasi exclusively into NHEJ[20]. In NHEJ-deficient (*Xrcc4*[-/-]) pro-B cells, RAG- or CRISPR/Cas9-DSBs that persist into subsequent S-G2/M cell cycle phases (due to p53-deficiency) are repaired through Pol θ-dependent mechanisms that generates chromosomal deletions and translocations[16]. CSR enables antibody isotype switching—from IgM to IgG, IgE, or IgA—while maintaining antigen specificity, thereby diversifying antibody functions and enhancing immune responses. CSR is initiated by activation-induced cytidine deaminase (AID), which deaminates cytosines in the transcriptionally active switch (S) regions of the immunoglobulin heavy chain (*Igh*) locus, converting them to uracils[21]. These S regions are G-rich repetitive sequences upstream of each constant coding segment, and through base excision and mismatch repair pathways, DNA breaks are induced between the donor Sμ region (upstream of IgM) and a recipient S region (e.g., Sγ, Sε, or Sα), corresponding to the new antibody isotype[21]. While NHEJ remains the primary repair mechanism employed to resolve AID-mediated G1-DSBs, studies have shown that class switching is only partially reduced (by approximately 60%) in B cells deficient in NHEJ components (e.g., Ku70, Ku80, XRCC4, or Ligase IV)[22–25], indicating that alt-EJ is proficient at repairing AID-induced DSBs when NHEJ is compromised. Indeed, in contrast to V(D)J recombination that is strictly confined to G0/G1-phase cells[20], CSR occurs in cycling cells, thus in a context which is more permissive to employ various pathways for DSB repair[14,26,27]. The nature of the alt-EJ pathway acting during CSR remains incompletely understood, and whether TMEJ acts as the primary repair mechanism is unknown[26,28]. During CSR, 53BP1 and RIF1 favors NHEJ by protecting AID-DSB ends against nucleolytic degradation through the recruitment of the REV7-SHLD1-SHLD2-SHLD3 (Shieldin) and CTC1–STN1–TEN1 (CST) complexes[28]. When NHEJ or DSB end-protection is impaired, the use of MH increases, indicating a shift toward alt-EJ[26]. NHEJ or DSB end-protection deficiencies are also associated with the accumulation of unproductive class switch joints that have suffered enhanced DSB end resection and/or inversional joining and, with the accumulation of chromosomal breaks at immunoglobulin (Ig) switching sites.

In this study, we address the involvement of Pol θ during CSR, particularly in NHEJ-deficient, *Xrcc4*[-/-], and Shieldin-deficient, *Shld1*[-/-], primary B cells. We show that Pol θ is dispensable to switch from IgM to IgG1/IgG2b/IgG3/IgA. However, in both NHEJ-deficient and SHLD unprotected cells, we find that Pol θ mediates the joining of resected and inversional AID- induced DNA-ends via MH-mediated end-joining. Furthermore, we question if, like in HR-deficiency, TMEJ occurs during mitosis in a RHINO/PLK1-dependent manner in NHEJ-deficient cells. We show that in *Xrcc4*[-/-] cells, both RAG- or AID-induced breaks are repaired via TMEJ in a RHINO-independent manner. Additionally, we demonstrate that TMEJ is active in G1/S phase and that it is not reliant on PLK1 activity, proposing a model where G1-induced breaks are repaired by TMEJ in S-phase and not in mitosis.

## Results

### Pol θ is dispensable for *Igh* CSR in wild-type, XRCC4-deficient and SHLD1-deficient B cells

In wild-type (WT) B cells, NHEJ is the major repair pathway implicated in AID-mediated break repair. Nonetheless, NHEJ-deficient cells sustain robust CSR via alt-EJ[22–25]. Although NHEJ proficient, Shieldin-complex-deficiencies (knock-out of *Mad2l2/Rev7*, *Shld1* or *Shld2*) result in severe CSR defects associated with exacerbated DSB end resection and MH usage[29–31]. Thus, we sought to investigate the role of Pol θ during CSR, in WT, NHEJ-deficient and Shieldin-unprotected splenic B cells. Using *Polq*[-/-] mice harboring the *Polq*[tm1Js] allele[32], we generated *CD21-Cre*[tg] *Xrcc4*[flox/flox] *Polq*[-/-] mice (Supplementary Fig. 1a), hereafter *Xrcc4*[-/-] *Polq*[-/-], by crossing *Polq*[-/-] mice with *CD21-Cre*[tg] *Xrcc4*[flox/flox] mice[24,33]. XRCC4 plays an imperative role in B cell development during V(D)J recombination, the first set of rearrangements occurring in B cell progenitors[34]. Hence, CD21 driven NHEJ-deficiency allowed us to obtain naïve mature B cells, capable to undergo CSR upon cytokine stimulations, and to circumvent embryonic lethality[24,25,34]. As expected, neither B nor T cell developments were disrupted in conditional *Xrcc4*[-/-] mice, regardless of the presence or absence of Pol θ (Supplementary Fig. 1b–i). Additionally, we crossed *Polq*[-/-] mice to *Shld1*[-/-] mice[29] generating *Shld1*[-/-] *Polq*[-/-] double deficient mice (Supplementary Fig. 2a). Similarly to *53bp1*[-/-] *Polq*[-/-] animals[35], *Shld1*[-/-] *Polq*[-/-] mice were viable and born at expected mendelian ratios when crossing *Shld1*[-/-] *Polq*[+/-] individuals together (Supplementary Fig. 2b) and total splenocytes were not statistically different across backgrounds, though *Shld1*[-/-] *Polq*[-/-] mice trended fewer than others (Supplementary Fig. 2c). Nevertheless, consistent total numbers of splenocytes and uniform CD19[+]IgM[+] cell population proportions were obtained in all harvested spleens (Supplementary Fig. 1b, i, Supplementary Fig. 2c-d) allowing efficient recovery of CD19[+] primary splenocytes and testing of their ability to undergo CSR upon isotype-specific cytokine stimulation.

As previously reported[36], *Polq*[-/-] B cells exhibited a comparable CSR efficiency at day 4 to WT B cells when prompted to transition from IgM-to-IgG1, average of 16.62 ± 1.52% in WT and 15.12 ± 1.94% in *Polq*[-/-] (Fig. 1a, b). Conversely, XRCC4 deficiency resulted in a mean 60% reduction in IgG1[+] cells following cytokine stimulation (Fig. 1a, b), as previously described[24,25]. Intriguingly, we observed that Pol θ is also dispensable for IgM-to-IgG1 CSR in B cells lacking XRCC4 by measuring mean CSR rate drops from WT (*Xrcc4*[-/-] 60% and *Xrcc4*[-/-] *Polq*[-/-] 68%) (Fig. 1a, b). Indeed, no statistical difference (Mann-Whitney, *p* = 0.355) in the IgG1[+] cells was obtained at day 4 post cytokine stimulation between *Xrcc4*[-/-] (6.25 ± 1.18%) and *Xrcc4*[-/-] *Polq*[-/-] (4.86 ± 0.90%). We also assessed IgM-to-IgG1 class switching kinetics in stimulated *Polq*[+/+] and *Polq*[-/-] B cells (± NHEJ), again non-significant differences were detected between Pol θ-proficient and Pol θ-deficient cells (Supplementary Fig. 1j). Furthermore, Pol θ 's dispensability extended beyond IgM-to-IgG1 switching, as cells efficiently recombined toward IgG2b, IgG3, and IgA isotypes in XRCC4-proficient (WT) cells (Fig. 1e–g).

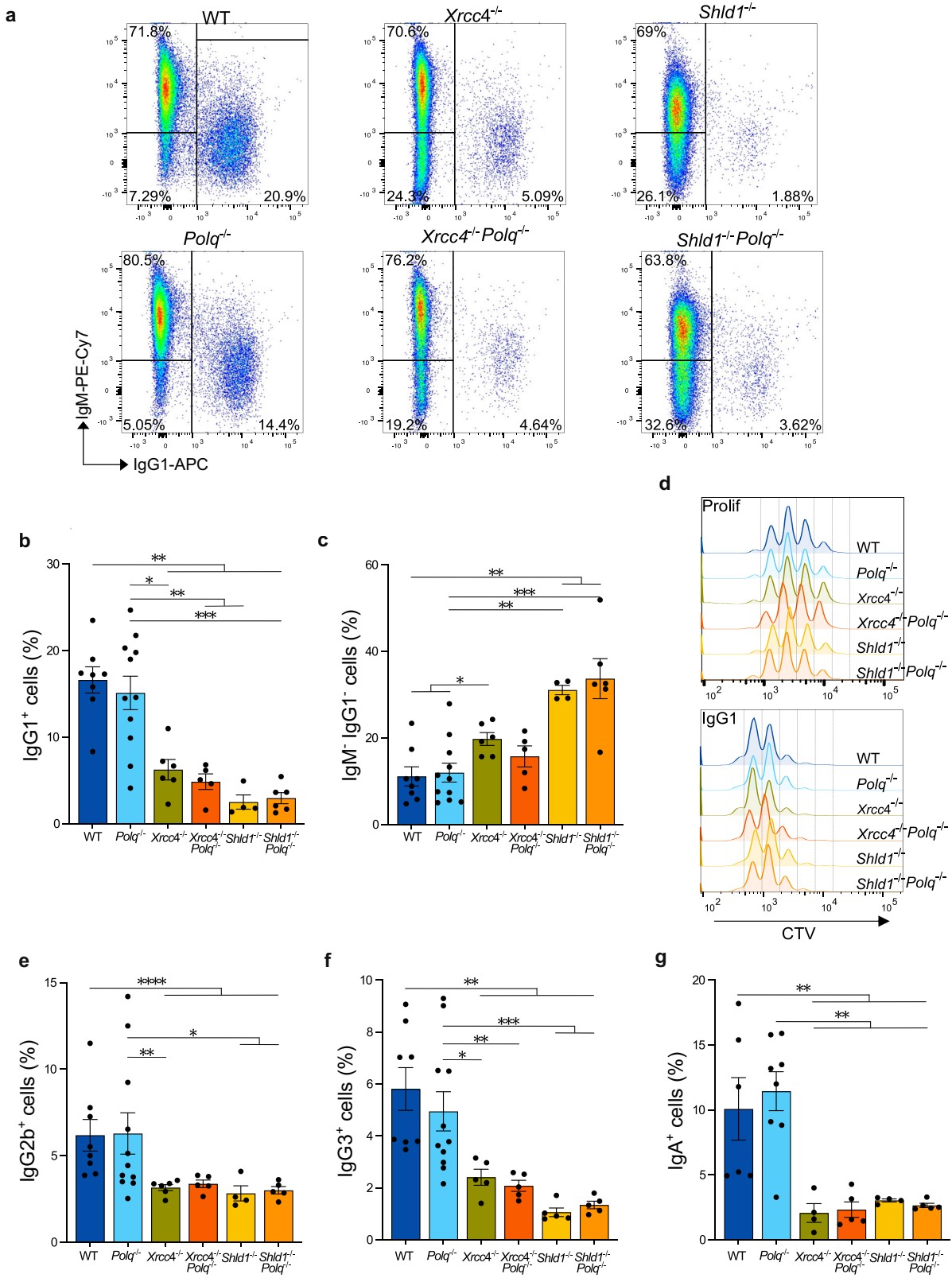

Although XRCC4-deficient B cells switch less efficiently towards IgG2b, IgG3, and IgA (compared to WT), Pol θ is also dispensable in this genetic setting (Fig. 1e-g). *Xrcc4^-/- Polq^-/-* immortalized progenitor B (pro-B) cells were shown to display a growth defect in cycling conditions[16]. In contrast to pro-B cells, *Xrcc4^-/-* and *Xrcc4^-/- Polq^-/-* CD19+ splenocytes displayed similar proliferation profiles when stained with CellTracer Violet (CTV) in both AID-induced and uninduced conditions (Fig. 1d). Likewise, non-significant differences in total cell counts were obtained at day 4 of the IgG1 CSR kinetic assay (Supplementary Fig. 1k), indicating that no striking growth defect is observed between genetic backgrounds. Despite successfully undergoing recombination and expressing a new switched isotype on their surface (*i.e.*, productive recombination), approximately 11% of WT and *Polq^-/-* CD19+ B cells became Ig-negative upon ex vivo CSR stimulation

**Fig. 1 | Pol θ is dispensable for productive CSR. a** Representative FACS plots of IgM and IgG1 expression in WT, *Polq⁻/⁻*, *Xrcc4⁻/⁻*, *Xrcc4⁻/⁻ Polq⁻/⁻*, *Shld1⁻/⁻* and *Shld1⁻/⁻ Polq⁻/⁻* primary B cells (CD19⁺), at day 4 post cytokine exposure (anti-IgD dextran/LPS/IL-4). **b, c** Quantifications of cells expressing IgG1⁺ (**b**) and of Ig-negative (IgM⁻ IgG1⁻) (**c**) cells. Bars represent mean ± SEM, Two-tailed Mann-Whitney, WT (*n* = 8), *Polq⁻/⁻* (n = 11), *Xrcc4⁻/⁻* (*n* = 6), *Xrcc4⁻/⁻ Polq⁻/⁻* (*n* = 5), *Shld1⁻/⁻* (*n* = 4) and *Shld1⁻/⁻ Polq⁻/⁻* (n = 6), *p* values are indicated in Source Data file. **d** Cell trace violet (CTV) profiles in proliferating (anti-IgD dextran) (top) and IgG1 stimulating (bottom) conditions at day 3. **e–g** Percentage of switched cells at day 4 post cytokine

exposure towards (**e**) IgG2b (anti-IgD dextran/LPS), WT (*n* = 8), *Polq⁻/⁻* (*n* = 11), *Xrcc4⁻/⁻* (*n* = 6), *Xrcc4⁻/⁻ Polq⁻/⁻* (*n* = 5), *Shld1⁻/⁻* (*n* = 4) and *Shld1⁻/⁻ Polq⁻/⁻* (*n* = 5), **f** IgG3 (anti-IgD dextran/LPS), WT (*n* = 8), *Polq⁻/⁻* (*n* = 11), *Xrcc4⁻/⁻* (*n* = 5), *Xrcc4⁻/⁻ Polq⁻/⁻* (*n* = 5), *Shld1⁻/⁻* (*n* = 5) and *Shld1⁻/⁻ Polq⁻/⁻* (*n* = 5) and **g** IgA (anti-IgD dextran/TGF-β/LPS/IL-5/RA), WT (*n* = 6), *Polq⁻/⁻* (*n* = 8), *Xrcc4⁻/⁻* (*n* = 4), *Xrcc4⁻/⁻ Polq⁻/⁻* (*n* = 5), *Shld1⁻/⁻* (*n* = 4) and *Shld1⁻/⁻ Polq⁻/⁻* (*n* = 5). Bars represent mean ± SEM, Two-tailed Mann-Whitney, *p* values are indicated in the Source Data file. Source data are provided as a Source Data file.

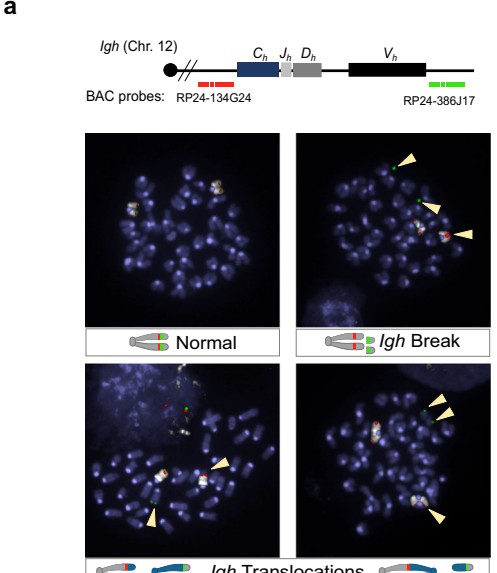

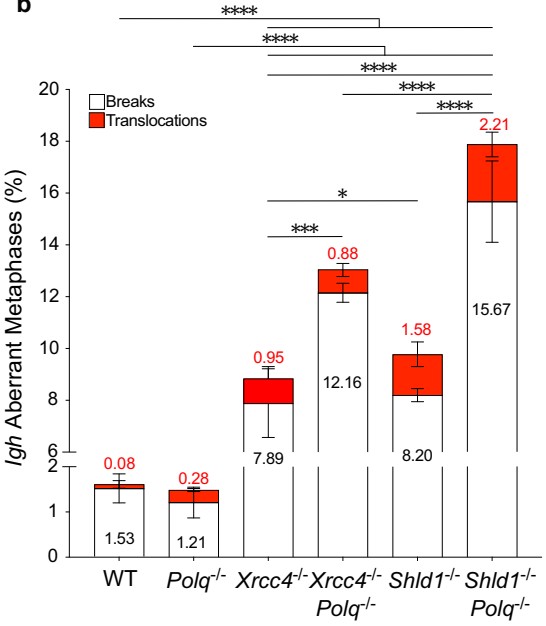

**Fig. 2 | Pol θ promotes *Igh* stability in NHEJ-deficient and SHLD-unprotected B cells. a** Scheme of DNA-FISH probes used to quantify *Igh* aberrant metaphases, break and translocation events, along with representative images. **b** Quantification of mean ( ± SEM) *Igh* breaks (white) and translocations (red) in metaphases spreads of day 4 IgG1 stimulated B cells, from 3-4 independent experiments. Total

metaphases analyzed: WT (*n* = 2044), *Polq⁻/⁻* (*n* = 2797), *Xrcc4⁻/⁻* (*n* = 1202), *Xrcc4⁻/⁻ Polq⁻/⁻* (*n* = 1083), *Shld1⁻/⁻* (*n* = 2025) and *Shld1⁻/⁻ Polq⁻/⁻* (*n* = 1829). A two-sided Fisher's Exact test was conducted using total numbers of normal and aberrant metaphases tabulated in each genotype, *p* values are indicated in Supplementary Data 1.

(Fig. 1a, c). CD19⁺ cells that no longer express IgM or IgG1 on their surface typically have undergone either exacerbated resection at DNA ends, inversional recombination events during CSR, or harbor unrepaired *Igh* breaks[29,30]. This Ig-negative population almost doubled to 19.73 ± 1.49% in *Xrcc4⁻/⁻* B cells and, remained nearly unchanged at 15.71 ± 2.43% upon Pol θ abolition (Fig. 1a, c). Thus, regardless of their XRCC4 status, *Polq⁻/⁻* B cells exhibit non-significant differences in IgG1⁺ and Ig-negative populations upon CSR, as compared to *Polq⁺/⁺* controls (Fig. 1a, c).

AID-mediated breaks are channeled into NHEJ repair via the 53BP1-RIF1-SHLD axis that limits end resection. We recapitulated severe CSR defect observed upon SHLD1 abolition[29], stimulated *Shld1⁻/⁻* B cells displayed a significant 85% decrease in productive CSR (IgM-to-IgG1) (Fig. 1a, b) and accumulated Ig-negative cells (Fig. 1a, c), approximately three-fold more as compared to WT and *Polq⁻/⁻* B cells. Equally to *Shld1⁻/⁻* B cells, 2.96 ± 0.64% of *Shld1⁻/⁻ Polq⁻/⁻* CD19⁺ purified splenocytes successfully switch from expressing IgM to IgG1 on their surface while, 33.67 ± 4.62% become Ig-negative at 4 days post cytokine stimulation (Fig.1a–c, Supplementary Fig. 2f). As observed in the XRCC4-deficient cells, knock-out of *Polq* in SHLD1-deficient B cells did not impact CTV proliferation profiles (Fig. 1d) nor final cell counts obtained during IgG1 CSR kinetic assay (Supplementary Fig. 2e, f). Overall, no significant differences in productive CSR towards IgG1 (Mann-Whitney, *p* = 0.7619), IgG2b (Mann-Whitney, *p* = 0.4127), IgG3 (Mann-Whitney,

*p* = 0.1508) and IgA (Mann-Whitney, *p* = 0.1905) were also obtained between SHLD1 and SHLD1 Pol θ double-deficient B cells (Fig. 1a, b, e–g). Hence, Pol θ is also dispensable for productive CSR when AID-mediated DNA-ends are unprotected by the Shieldin-complex.

## TMEJ joins a subset of AID-induced DNA breaks during CSR

Although we show that Pol θ is dispensable for productive CSR, in WT, XRCC4-deficient and SHLD1-deficient backgrounds, we further investigated the non-productive CSR events. As stated above, the Ig-negative population, a proxy for unproductive CSR events using FACS analysis, is composed of several recombination outcomes, including unrepaired *Igh* breaks, highly resected joints and inversions. An increased number of *Igh*-unrepaired breaks and *Igh*-mediated translocations have been previously described in XRCC4-deficient and SHLD-unprotected B cells[24,29,37]. Thus, we next quantified both these types of events in WT, *Polq⁻/⁻*, *Xrcc4⁻/⁻*, *Xrcc4⁻/⁻ Polq⁻/⁻*, *Shld1⁻/⁻* and *Shld1⁻/⁻ Polq⁻/⁻* stimulated primary splenic B cells. Using BAC-probes hybridizing at each end of mouse *Igh* locus, we quantified aberrant metaphases containing *Igh*-specific breaks and translocations events (Fig. 2a). To carry out this analysis in the most comprehensive way, we analyzed thousands of metaphases generated from independent B-cell stimulations (Supplementary Data 1). WT and *Polq⁻/⁻* B cells stimulated for IgG1 class switching displayed roughly equal mean numbers of aberrant metaphases, 1.61 ± 0.29% and 1.49 ± 0.35%, respectively.

However, as previously reported[36], a 3.27-fold increase in translocations is obtained in $Polq^{-/-}$ B cells as compared to WT (Fig. 2b, Supplementary Data 1). Additionally, $Xrcc4^{-/-}$ splenocytes possess 7.89 ± 1.33% $Igh$-breaks and 0.95 ± 0.46 % of $Igh$ translocations, roughly 5.49x and 5.95x more aberrant metaphases than WT and $Polq^{-/-}$ B cells, respectively (Fig. 2b, Supplementary Data 1). $Shld1^{-/-}$ B cells also accumulate $Igh$-chromosomal aberrations, 8.2 ± 0.25% of breaks and 1.58 ± 0.48 % of translocations (Fig. 2b, Supplementary Data 1). Notably, although productive and unproductive CSR events are unchanged upon $Polq$-deletion via FACS analysis (Fig. 1), we observe significantly more aberrant metaphases in $Xrcc4^{-/-} Polq^{-/-}$ (13.04 ± 0.11% aberrant metaphases) and $Shld1^{-/-} Polq^{-/-}$ (17.88 ± 1.90% aberrant metaphases) cells, when comparing to single-deficiency counterparts $Xrcc4^{-/-}$ (8.84 ± 1.66% aberrant metaphases) or $Shld1^{-/-}$ (9.78 ± 0.43% aberrant metaphases) B cells (Fig. 2b, Supplementary Data 1), implying that Pol θ is involved in repairing a subset of AID-induced $Igh$ breaks, particularly in XRCC4-inactive or SHLD1-unprotected contexts.

## TMEJ generates resected and inversional unproductive joints during CSR

To identify the type of DNA-end joined by Pol θ in CSR, we first employed a long-range PCR (LR-PCR) assay to amplify a region between the $Igh$ intronic enhancer iEμ (primer EμF) and the $Igh$ γ1 constant exon 6 (primer Cγ1R2)[29] (Fig. 3a). This assay amplifies in bulk all IgM-to-IgG1 deletional recombined events within a sample, as primers used are too far apart on the germline locus (>100 kb apart) (Fig. 3a). Productive recombination events that occurred strictly between Sμ and Sγ1 (S-S joints) range in size of 4.9–17.3 kb, while extensively resected CSR joints exiting both S-region loci are readily discernible via LR-PCR as they produce shorter amplicons (resected joints, ≤4.9 kb) (Fig. 3a). In line with FACS acquired CSR-assays, LR-PCR fragment length profiles, amplified from IgG1-stimulated cells, revealed that most recombination events occur within S-regions in all tested B cell genetics, generating 4.9–17.3 kb S-S joint fragments (Fig. 3b, Supplementary Fig. 3a). However, upon XRCC4-inactivation, an increased number of shorter fragments (smear) appears below the S-S joints (Supplementary Fig. 3a), consistent with previous reports showing that the number of resected $Igh$ joints increases upon abolition of NHEJ[29,38,39]. Remarkably, $Xrcc4^{-/-} Polq^{-/-}$ samples lack a visible smear on agarose gel suggesting that resected joints are not formed in the absence of Pol θ (Supplementary Fig. 3a). In contexts where DNA-end resection is exacerbated, such as SHLD1-deficiency, an expected weaker S-S joints band and an abundant smear of resected joints below it are obtained by LR-PCR (Fig. 3b, Supplementary Fig. 3a) that is consistent with reduced IgG1+ and increased Ig-negative population percentages in these cells[29] (Fig.1a, c). In this setting, the effect of Pol θ-deletion reveals a crisp S-S joints band, and no smear (resected joints) (Fig. 3b, Supplementary Fig. 3a). To precisely quantify amplified S-S and resected $Igh$ joints, we sequenced LR-PCR fragments using PacBio long-read sequencing technology from 3–6 independent B cell samples per genotype. Read length distributions recapitulate LR-PCR migration profiles (Fig. 3c, Supplementary Fig. 3b) as each long-read represents an IgM-to-IgG1 deletional recombination event. Using tens-of-thousands of reads per genetic background (Supplementary Data 2), median read lengths of 8.9 and 9.1 kb is calculated from WT and Pol θ-deficient cells, respectively, of which 1.76 and 0.87% of reads are below the threshold of productive CSR (≤4.9 kb) (Fig. 3c, Supplementary Fig. 3b, Supplementary Data 2). Consistent with the observed exacerbated DNA-end resected joints from agarose gels, the median deletion size significantly drops further by 0.9–1.2 kb in $Xrcc4^{-/-}$ and by 2.5–2.8 kb in $Shld1^{-/-}$ stimulated B cells and 11.5% ($Xrcc4^{-/-}$) and 26.8% ($Shld1^{-/-}$) of total reads are unproductive (Fig. 3c, Supplementary Fig. 3b, Supplementary Data 2). Strikingly, recovery back to WT-like deletion sizes occurs with added Pol θ deficiency ($Xrcc4^{-/-} Polq^{-/-}$ median = 9.2 kb, 1.58% ≤4.9 kb and $Shld1^{-/-} Polq^{-/-}$ median = 10.2 kb,

0.154% ≤4.9 kb) (Fig. 3c, Supplementary Fig. 3b, Supplementary Data 2). Mapping unique junction positions from Sμ and Sγ1 deletions (Supplementary Data 3) revealed less than 1.03% of unique break joining events are located outside S-regions in both NHEJ-deficient and SHLD-unprotected contexts without Pol θ (Fig. 3d, e), which is within the range of WT and $Polq^{-/-}$ backgrounds (0.47–2.5%) and well below $Xrcc4^{-/-}$ (13.04%, 5.2x of WT) and $Shld1^{-/-}$ (19.68%, 7.9x of WT) cells (Fig. 3d, e, Supplementary Data 3). Overall, via LR-PCR sequencing, we show that the repair of resected deletional $Igh$ joints, resulting in unproductive CSR, is mediated exclusively by Pol θ.

In NHEJ-deficient and SHLD-unprotected cells, increased inversional recombination events occur upon joining of the wrong DSB-ends together, potentially a consequence of extensive DNA-end processing resulting in synapsis destabilization[29]. Inversional recombination is the third type of CSR event leading to the Ig-negative phenotype. As LR-PCR sequencing is limited to the analysis of deletional events, we also examined the pattern of Sμ-Sγ1 joints by high-throughput genome-wide translocation sequencing (HTGTS) using gDNA from IgG1-stimulated cells[29,40]. A single bait primer annealing at the 5′ of the Sμ region is used to linearly amplify the junctional sequences and thus, identify and characterize the downstream S-region DSB joined to the donor Sμ (Fig. 4a). This high-throughput CSR joints sequencing approach permits the characterization of both deletional (blue arrows) and inversional (red arrows) CSR events occurring intra-Sμ (within the Sμ) and inter-S regions (e.g., Sμ-Sγ1) (Fig. 4a)[29,40,41]. In WT and $Polq^{-/-}$, no differences are observed as we identified 16,942 and 18,147 $Igh$ joints fusing the Sμ bait break to another Sμ or a downstream S break (Fig. 4b). Amongst these joints, just over 34% are derived from Sμ-Sμ joints and approximately 60% of $Igh$ joints are Sμ-Sγ1 fusion, while very few map to other S regions (i.e., Sγ3 and Sε) (Fig. 4b, Supplementary Fig. 4a, b). A more detailed investigation of the of Sμ-Sγ1 fusion distributions (WT 10,215 joints and $Polq^{-/-}$ 10,750 joints) show that the vast majority map within the Sγ1 and that >95% of joints are deletional with few inversional (<5%) recombination events (Fig. 4b). Consequently, less than 1% of Sμ-Sγ1 joints are resected (Fig. 4c) with an average INV/DEL ratio of 0.04 ± 0.01 for WT and $Polq^{-/-}$ IgG1-stimulated cells (Fig. 4d). Deletion of XRCC4 or SHLD1 remarkably diminishes by half the total number of $Igh$ joints sequenced (9094 joints $Xrcc4^{-/-}$, 8464 joints $Shld1^{-/-}$) (Fig. 4b) and significantly decreases the proportion of Sμ-Sγ1 (<45%) (Fig. 4b, Supplementary Fig. 4a, b), as previously described[23,29,38]. In $Xrcc4^{-/-}$ we recapitulate the increasing Sμ-Sμ joints[23,29,38] while no significant difference is observed upon SHLD1-depletion as compared to WT (Supplementary Fig. 4a). Zooming in on the Sμ-Sγ1 join distributions ($Xrcc4^{-/-}$ 3631 joints and $Shld1^{-/-}$ 2261 joints) confirms that although the majority of joints are deletional events (88% in $Xrcc4^{-/-}$ and 77% in $Shld1^{-/-}$) an increased portion of CSR joints underwent inversional repair (12% in $Xrcc4^{-/-}$ and 23% in $Shld1^{-/-}$) as compared to WT, significantly boosting the average INV/DEL ratios to 0.116 ± 0.011 in $Xrcc4^{-/-}$ and to 0.257 ± 0.037 in $Shld1^{-/-}$ B cells (Fig. 4b, d). Independently of orientation, the number of Sγ1 fusions mapping outside Sγ1, resected CSR joints, are increased by 12.9- and 39.7-fold in $Xrcc4^{-/-}$ and $Shld1^{-/-}$, respectively (Fig. 4b, c). In NHEJ-deficient and SHLD1-unprotected cells, the absence of Pol θ plumets the total number of $Igh$ joints to ~25% from single knockouts and <12% of WT (Fig. 4b), corroborating the accumulation of unrepaired AID-breaks observed in metaphase spreads (Fig. 2). Despite the significant drop in recovered junctions, added Pol θ-deficiency surprisingly enriches S region junctions by specifically eliminating the resected joints flanking the S regions (Fig. 4b, c). A focus on Sμ-Sγ1 joints reveals that in both double-deficient models virtually no prey sequences map outside the Sγ1 region (average of 0.072 ± 0.072% joints in $Xrcc4^{-/-} Polq^{-/-}$ and 0.446 ± 0.446% joints in $Shld1^{-/-} Polq^{-/-}$) (Fig. 4b, c), as observed using the LR-PCR sequencing approach. Furthermore, Pol θ-deficiency

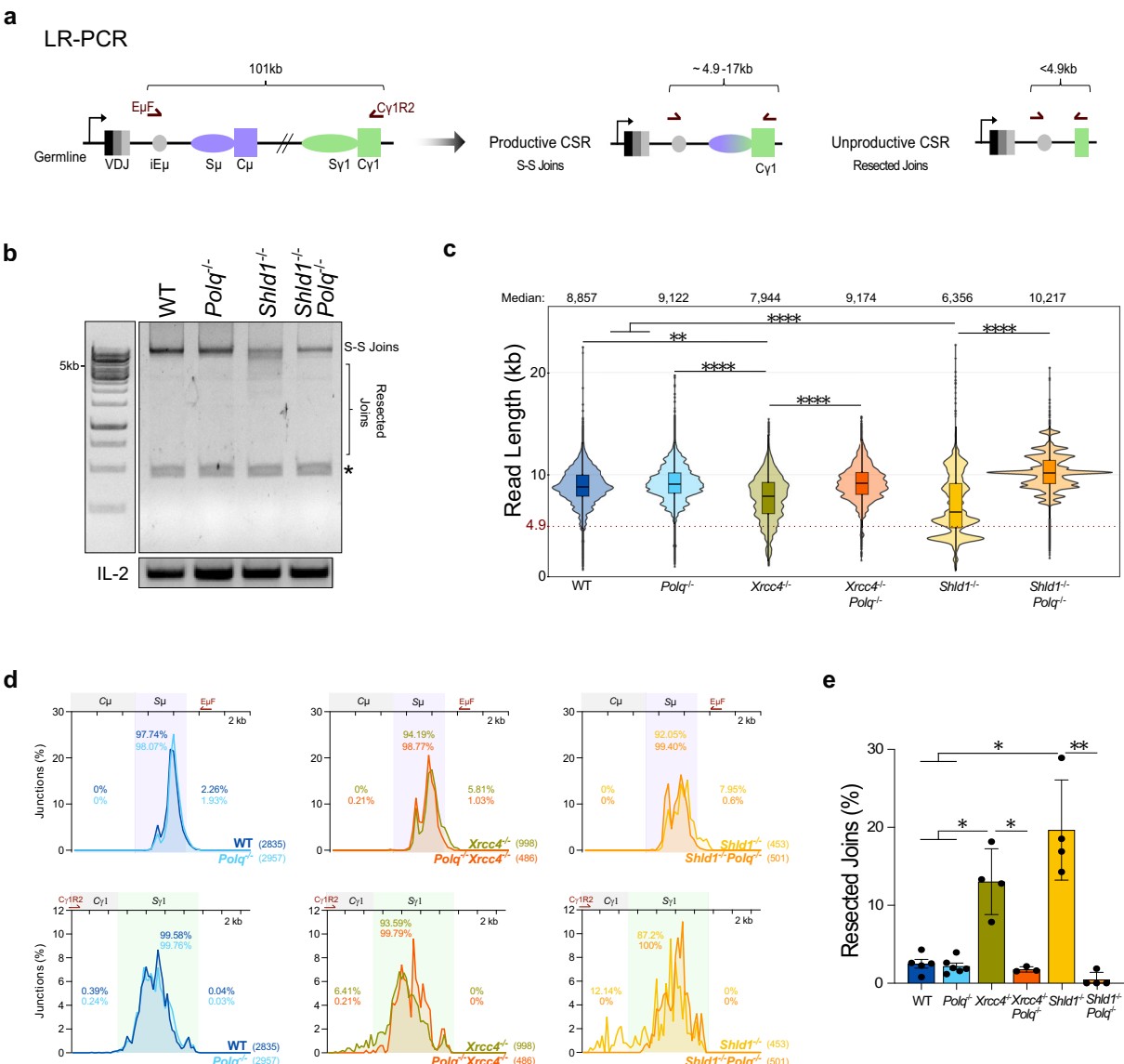

**Fig. 3 | TMEJ is responsible for the repair (long-)resected AID-induced recombination events.** Using primary B cells (CD19 +) day 4 post stimulation with anti-IgD dextran/LPS/IL-4. **a** Schematic long-range PCR (LR-PCR) visualizing resection tracts during IgG1 switching. Primers (burgundy arrows) in intronic enhancer iEμ and Ighy1 exon 6, amplify deletional rearrangements. Productive Sμ-Sγ1 joins yield 4.9–17.3 kb amplicons (S-S joints); resected, unproductive events yield fragments <4.9 kb. **b** LR-PCR profiles of Sμ-Sγ1 junctions. IL-2 is a loading control. See Supplementary Fig. 3. * indicates nonspecific band. **c** Violin plots of read lengths obtained by PacBio sequencing of LR-PCR fragments, from independent samples WT (*n* = 5), *Polq*−/− (*n* = 6), *Xrcc4*−/− *Polq*−/− (*n* = 3), *Xrcc4*−/−, *Shld1*−/− and *Shld1*−/− *Polq*−/− (*n* = 4). Mixed-model regression test used followed by post-hoc Tukey HSD test, WT vs. *Xrcc4*−/− (*p* = 0.0027), WT vs. *Shld1*−/− (*p* = 1.11e-08), *Polq*−/− vs. *Xrcc4*−/− (p = 7.46e-05), *Polq*−/− vs. *Shld1*−/− (p = 2.00e-11), *Xrcc4*−/− vs. *Xrcc4*−/−*Polq*−/− (*p* = 0.00053) and *Shld1*−/− vs. *Shld1*−/− *Polq*−/− (*p* = 5.65e-14). For all boxplots: minima is minimum value, maxima is maximum value, center is median and quartiles shown by box and whiskers. See Supplementary Data 2 for number of reads plotted. **d** Unique breakpoints obtained via LR-PCRseq, were pooled by genotype and distribution of AID-junctions situated at Sμ and Sγ1 regions of WT, *Polq*−/−, *Xrcc4*−/−,*Xrcc4*−/− *Polq*−/−, *Shld1*−/− and *Shld1*−/− *Polq*−/− cells was plotted. Coordinates of unique breakpoints were used to generate distributions and to determine the percentage of unique junctions across Sμ and Sγ1 genomic regions, using 250 bp bins. Percentage of unique junctions located within and around Sμ and Sγ1 genomic regions are indicated. The total number of plotted junctions (n) is specified between brackets in the bottom right corner of each graph. Primers used for LR-PCR are in dark red. **e** Percentage of resected joints, break junction located outside a S-region, obtained via LR-PCRseq using PacBio. Resected joints are defined as joints where either or both 5′ or/and 3′ coordinates are outside Sμ or/and Sγ1 loci, amongst unique junctions. Bars represent mean ± SEM, Unpaired two-tailed T-test (*p < 0.05, **p < 0.01). See also Supplementary Data 3. Source data are provided as a Source Data file.

reduces the inversion proportion within S regions, resulting in a restored INV/DEL ratio (0.05 ± 0.021 and 0.061 ± 0.016, in *Xrcc4*−/− *Polq*−/− and *Shld1*−/− *Polq*−/−) (Fig. 4d). Thus, aberrant non-productive CSR joints resulting from exacerbated resection and/or inversional repair are solely mediated by Pol θ-dependent repair pathway.

Several studies have shown that, in absence of NHEJ, CSR breakpoints exhibit increased short MH usage thus it is generally thought the alt-EJ pathway(s) backing up CSR is/are MH-driven[22,24,29,38,39,42]. Notably, Pol θ-dependent repair pathway requires resection and exposure of short MH tracts at DSBs, subsequently used to tether

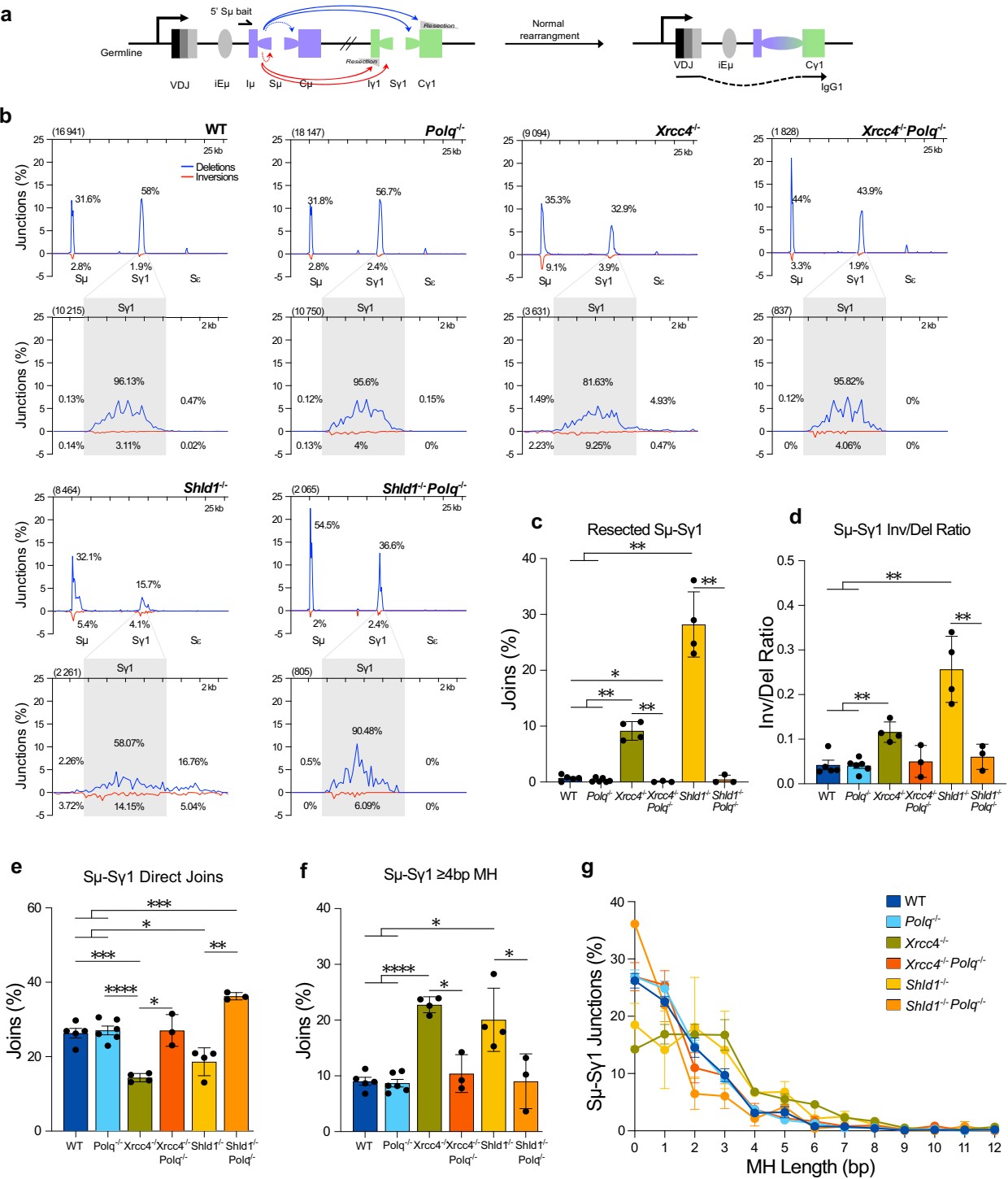

together DNA-ends for repair. Thus, the Pol θ repair signature is associated with short MH (4–8 bp) usage and limited >1 bp insertions[43]. Next, we leveraged our HTGTS CSR data to analyze breakpoint repair signatures, such as short insertions and MH, in stimulated primary splenic B cells and address the impact of Pol θ-absence on CSR junction types. The following analyses are conducted on Sμ-Sγ1 joints, HTGTS prey sequences mapping to the Sγ1 ± 10 kb. We first focused on quantifying direct joints, joints deprived of MH or insertions. In WT and *Polq*[-/-] IgG1 stimulated B cells, roughly 26% of Sμ-Sγ1 joints are direct joints and 9% of joints possess MH ≥ 4 bp, consistent with NHEJ being the major repair mechanism of AID-breaks during CSR (Fig. 4e-g). However, unlike previously reported[36], we observe no difference

between WT and *Polq*[-/-] cells with regards to >1 bp insertions (Supplementary Fig. 4c, d), potentially stemming from the employment of distinct CSR sequencing techniques. As expected[22,24,29,38,39,42], direct joints significantly drop by 1.84-fold to an average of 14.23% in NHEJ-deficient (*Xrcc4*[-/-]) and by 1.42-fold to an average of 18.5% in SHLD-unprotected (*Shld1*[-/-]) B cells (Fig. 4e, g) while percentages of Sμ-Sγ1 joints possessing short MHs increases (Fig. 4f, g). Plotting of MH usage landscapes revealed an increase of joints harboring 4–8 bp long MH in *Xrcc4*[-/-]- and *Shld1*[-/-]-single deficient B cells (Fig. 4g). Indeed, percentages of sequenced Sμ-Sγ1 joints with MH ≥ 4 bp long significantly increase by 2.52-fold in *Xrcc4*[-/-] (22.75%) and by 2.22-fold in *Shld1*[-/-] (20.06%) (Fig. 4f, g). Markedly, in both NHEJ-deficient and

**Fig. 4 | CSR-HTGTS reveals that TMEJ repairs unproductive (long-)resected and inversional AID-induced recombination events. a** Schematic representation of activation of CSR in normal B cells stimulated with anti-IgD dextran/LPS/IL-4 which induces AID and activates transcription of *Iγ1*. Deletional CSR events are indicated with a blue arrow and inversional recombination events are indicated with a red arrow. Dotted line indicates intra-switch recombination events. The bait primer used for CSR- HTGTS is situated upstream of Sμ region (**b**) CSR-HTGTS-seq analysis of break points between 5'Sμ and downstream acceptor S-regions in WT, *Polq⁻/⁻*, *Xrcc4⁻/⁻*, *Xrcc4⁻/⁻ Polq⁻/⁻*, *Shld1⁻/⁻* and *Shld1⁻/⁻ Polq⁻/⁻* splenocytes stimulated with anti-IgD dextran/LPS/IL-4. Junctions obtained via CSR-HTGTS-seq conducted on 3–6 mice, were pooled by genotype and the distribution of total junctions was plotted in 1 kb bins across the *Igh*. Percentages of mapped deletional and inversional joints located within Sμ and Sγ1 are indicated on each distribution plot. To each plot, a zoom into the distribution of AID-junctions situated at Sγ1 region, was plotted in 250 bp bins. The total number of plotted junctions (n) is indicated between brackets in the top left corner of each graph. Percentages of mapped deletional and inversional joints within and outside of Sγ1 is indicated on each distribution plot. **c, d** Bars representing mean ± SEM, WT (n = 5), *Polq⁻/⁻* (n = 6), *Xrcc4⁻/⁻* (n = 4), *Xrcc4⁻/⁻ Polq⁻/⁻* (n = 3), *Shld1⁻/⁻* (n = 4) and *Shld1⁻/⁻ Polq⁻/⁻* (n = 3), Unpaired two-tailed T-test, *p* values are indicated in Source Data file. Percentage of resected Sμ-Sγ1 joints in **c**. Ratios of inversional versus deletional events for each sample in (**d**). CSR-HTGTS-seq analysis of repair signature of Sμ- Sγ1 joints of splenocytes stimulated with anti-IgD dextran/LPS/IL-4. **e–g** Bars representing mean ± SEM, WT (n = 5), *Polq⁻/⁻* (n = 6), *Xrcc4⁻/⁻* (n = 4), *Xrcc4⁻/⁻ Polq⁻/⁻* (n = 3), *Shld1⁻/⁻* (n = 4) and *Shld1⁻/⁻ Polq⁻/⁻* (n = 3), Unpaired two-tailed T-test, *p* values are indicated in Source Data file. Percentage of direct joints (**e**). Percentage of joints containing MH tracks of ≥ 4 bp at break site (**f**). **g** MH usage landscapes in each genetic background. Each data point represents the mean ± SEM, WT (n = 5), *Polq⁻/⁻* (n = 6), *Xrcc4⁻/⁻* (n = 4), *Xrcc4⁻/⁻ Polq⁻/⁻* (n = 3), *Shld1⁻/⁻* (n = 4) and *Shld1⁻/⁻ Polq⁻/⁻* (n = 3). Source data are provided as a Source Data file.

SHLD1-unprotected settings Pol θ-depletion restores the WT-phenotype, causing significant decreases of joints with short MH and significant increases of direct Sμ-Sγ1 joints (Fig. 4e-g). Hence, inversional and resected CSR joints are repaired in a Pol θ-dependent manner, favoring usage of short MH-tracts. In *Xrcc4⁻/⁻ Polq⁻/⁻* B cells, captured Sμ-Sγ1 joints represent only productive CSR events resulting from NHEJ/Pol θ-independent repair of AID-breaks, and use no or minimal MH (MH 0–2 bp) (Fig. 4g). Thus, Pol θ-independent end-joining is responsible for productive CSR in absence of NHEJ that is not reliant on MH usage while TMEJ generates MH-mediated non-productive joint formation.

### TMEJ of AID-induced DSBs does not require RHINO

Next, we asked if, like S/G2-induced breaks, G1-induced breaks are repaired by TMEJ in mitosis, in a RHINO dependent manner. To this aim, we engineered several single and double mutant CH12F3 B cell lines using CRISPR/Cas9-mediated gene editing (See Supplementary Data 8-9 for complete list of cell lines and primers used for genotyping). First, we abrogated NHEJ (XRCC4 ΔEx3) and/or Pol θ (ΔEx1) to obtain *Polq⁻/⁻*, *Xrcc4⁻/⁻* and *Xrcc4⁻/⁻ Polq⁻/⁻* B cell lines (Supplementary Fig. 5a, Supplementary Data 8). CH12F3 B cells behave like primary B cells, indeed the abolition of both NHEJ and Pol θ is dispensable for productive IgM-to-IgA switching levels (Fig. 5a, b) though leads to a significant accumulation of *Igh* associated breaks (Fig. 5c), due to the lack of joining of resected ends as visualized via LR-PCR (Fig. 5d and Supplementary Fig. 5c) and measured by LR-PCR sequencing (Supplementary Fig. 5d, e). We also confirmed the absence of resected CSR joins, along with corresponding increase of *Igh* breaks, upon treatment with the Pol θ inhibitor (Polθi) ART558 (Supplementary Fig. 6). Globally, we find that primary B cells and in vitro cell lines lacking NHEJ repair a subset of AID-mediated breaks via TMEJ.

We then constructed several independent RHINO-deficient ( ± NHEJ) CH12F3 B cells by targeting a deletion in exon 3, resulting in a frameshift and removal of the ATG[13] (Supplementary Fig. 7a-b). Upon cytokine stimulation (Fig. 5a), the abolition of Pol θ or RHINO does not impact IgM-to-IgA CSR as compared to WT, unlike XRCC4-depletion which leads to a significant 2-fold drop in IgA⁺ and a 3-fold increase in Ig-negative populations (Fig. 5b). Productive class switching is also unaltered in NHEJ-deficient cells, as *Xrcc4⁻/⁻ Rhno1⁻/⁻* or *Xrcc4⁻/⁻ Polq⁻/⁻* display non-significant differences to *Xrcc4⁻/⁻* single mutant cell lines (Fig. 5b). Thus, we established that RHINO is dispensable for IgM-to-IgA class switching in XRCC4-proficient and XRCC4-deficient cells (Fig. 5a, b). We then posed if *Rhno1⁻/⁻* deficient CH12F3 B cells behave differently to WT or NHEJ-deficient counterparts when analyzing *Igh* stability and joining products. We observe that RHINO-null clones display comparable levels of *Igh* instability (2.4 ± 0.35% breaks and 2.92 ± 0.11% translocations) to WT (1.27 ± 0.31% breaks and 0.29 ± 0.71% translocations) and *Polq⁻/⁻* (2.25 ± 0.72% breaks and 0.5 ± 0.19% translocations) stimulated CH12F3

clones (Fig. 5c, Supplementary Data 6). In NHEJ-deficient cells, the percentage of metaphases with *Igh* breaks strikingly increases by roughly two-fold in *Xrcc4⁻/⁻ Polq⁻/⁻* CH12F3 B cells (28.86 ± 2.02% breaks) as compared to XRCC4-deletion (14.41 ± 1.02% breaks) (Fig. 5c, Supplementary Data 6) while the combined loss of XRCC4 and RHINO only slightly increases *Igh* instability (18.42 ± 2.89% breaks and 2.92 ± 0.63% translocations) (Fig. 5c, Supplementary Data 6). AID-breaks repaired by Pol θ (*i.e.* those undergoing resection prior to joining) were readily detectable by LR-PCR in XRCC4/RHINO-double deficient cells (Fig. 5d), unlike in *Xrcc4⁻/⁻ Polq⁻/⁻* cells. Indeed, smear profiles of *Xrcc4⁻/⁻* and *Xrcc4⁻/⁻ Rhno1⁻/⁻* are undistinguishable in all independent clones tested. Overall, these results show that RHINO is dispensable for TMEJ of AID-induced DSBs in XRCC4-deficient B cells.

### TMEJ of RAG-induced DSBs occurs in G1/S-phase cells independently of RHINO and PLK1

We previously showed that in absence of NHEJ (*p53⁻/⁻ Xrcc4⁻/⁻* pro-B cells), G0/G1-induced RAG-DSBs are repaired by Pol θ upon release into subsequent cell cycle phases, S-G2/M[16]. However, it remains unclear when repair is specifically conducted, S-, S/G2- or M-phase. Thus, we interrogated if TMEJ of these breaks occurs during mitosis. We constructed multiple *p53⁻/⁻ Xrcc4⁻/⁻ Rhno1⁻/⁻*-triple deficient Abelson kinase (*v-Abl*) transformed BCL2-expressing pro-B cell clones using the same strategy mentioned above (Supplementary Fig. 7a–c). Then, we blocked *v-Abl* pro-B cells in G0/G1 through treatment with *v-Abl* kinase inhibitor (STI571, hereafter referred to as ABLki or STI). G0/G1 cell cycle arrest of *v-Abl* pro-B cells, rapidly leads to induction/stabilization of RAG1/2 gene expression and rearrangement of the endogenous *Igk* locus (Fig. 6a)[16,44]. In line with previous work[16], PCR amplification of endogenous *Igk* V₁₀₋₉₅-J₄ rearrangement revealed robust recombination in NHEJ-proficient p53-null cells, while all NHEJ-deficient pro-B cell clones completely lack coding joint formation in ABLki-treated condition (Fig. 6b). A substantial rescue of recombination in released/cycling *p53⁻/⁻ Xrcc4⁻/⁻* pro-B cells is obtained in the form of PCR amplified DNA products of highly variable sizes (Fig. 6b). In absence of Pol θ, no *Igk* V₁₀₋₉₅-J₄ rearrangements can be amplified from *p53⁻/⁻ Xrcc4⁻/⁻ Polq⁻/⁻ v-Abl* pro-B cells (Fig. 6b), demonstrating that unrepaired-RAG breaks exiting G1-phase are fully reliant on TMEJ. *Igk* genomic instability is increased by 20-fold in *p53⁻/⁻ Xrcc4⁻/⁻* pro-B (44.76 ± 16.28% of metaphases), as compared to *p53⁻/⁻* NHEJ-proficient pro-B (2.18 % aberrant metaphases) (Fig. 6c, Supplementary Data 7). Roughly half of the aberrant metaphases in *p53⁻/⁻ Xrcc4⁻/⁻* pro-B contained *Igk* translocations, that is 22.23 ± 7.89 % of total metaphases (Fig. 6c, Supplementary Data 7). In contrast, *p53⁻/⁻ Xrcc4⁻/⁻ Polq⁻/⁻* cells display virtually no *Igk* translocations (0.44% of metaphases) and instead accumulate high levels of *Igk* breaks (64.1% of metaphases) (Fig. 6c, Supplementary Data 7). We then addressed if RHINO is involved in the repair of RAG-mediated DSBs by evaluating *Igk* V₁₀₋₉₅-J₄

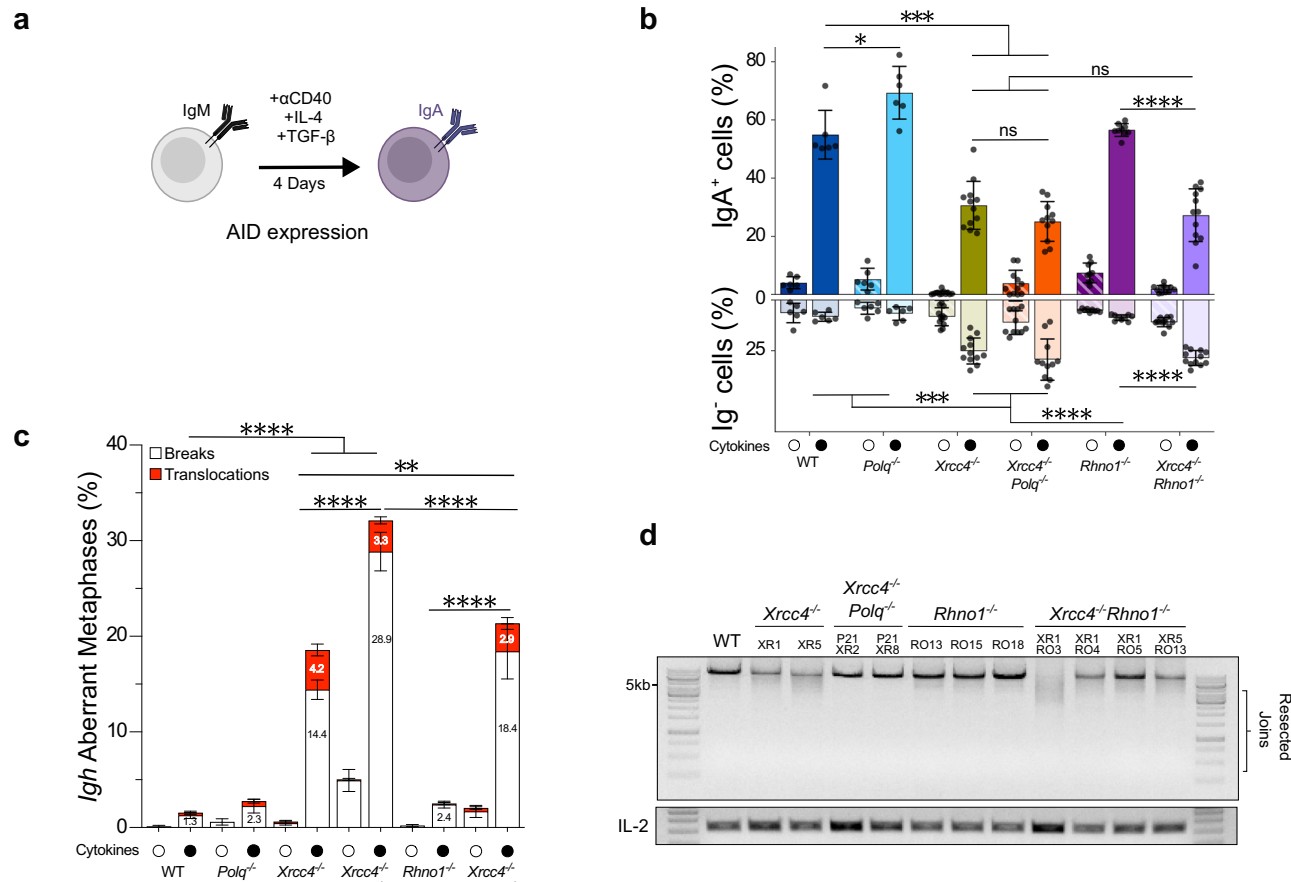

**Fig. 5 | AID-induced DSBs are repaired via TMEJ, independently of RHINO.**
**a** CH12F3 undergo IgM-to-IgA CSR upon in vitro stimulation with anti-CD40/ TGF-β/
IL-4. **b** Two bars for each genotype represent percentage of productive IgA⁺ and
unproductive Ig^neg CH12F3 cells at day 4, in untreated (striped bars) and cytokine
stimulated (full bars) conditions. Cell lines used: WT (CH12F3, n = 6), *Polq*^−/−
(POLQ13, n = 3; POLQ21, n = 3), *Xrcc4*^−/− (XR1, n = 6; XR5, n = 6), *Xrcc4*^−/− *Polq*^−/−
(POLQ21XR2, n = 5; POLQ21XR8, n = 6), *Rhno1*^−/− (RO13, n = 3; RO15, n = 3; RO18,
n = 3) and *Xrcc4*^−/− *Rhno1*^−/− (XR1RO3, n = 3; XR1RO4, n = 3; XR1RO5, n = 3; XR5RO13,
n = 3). Bars represent mean ± SEM, Two-tailed Mann-Whitney test, *p* values are
indicated in Source Data file. **c** Quantification of aberrant *Igh* metaphases, in
unstimulated or IgM-to-IgA stimulated CH12F3 cells. Cells were harvested at day 4
from 1-3 independent experiments with a minimum of 2 independent CRISPR-
clones per genotype. Histograms represent mean percentages of metaphases with
translocations (red) and chromosomes breaks (white) ± SEM. The mean percentage
of metaphases bearing breaks and translocations is indicated within bars. Total

metaphases analyzed in unstimulated and stimulated conditions, respectively: WT
(n = 799, n = 826), *Polq*^−/− (n = 803, n = 488), *Xrcc4*^−/− (n = 1 237, n = 1 595), *Xrcc4*^−/−
*Polq*^−/− (n = 618, n = 824), *Rhno1*^−/− (n = 963, n = 643) and *Xrcc4*^−/− *Rhno1*^−/− (n = 1071,
n = 852). A two-sided Fisher's Exact test was conducted using total numbers of
normal and aberrant metaphases tabulated in each genotypes, *p* values are indi-
cated in Supplementary Data 6. Also See Supplementary Data 6. **d** LR-PCR of IgM-
to-IgA rearrangements, using primers mapping within the intronic enhancer iEµ
and the *Igh* α constant exon 4 sequence. Productive CSR repair events occurring
strictly between Sµ and Sα result in amplicons spanning 5.1 kb-13.5 kb (S-S joints),
while unproductive CSR recombination events which underwent resection existing
both S-regions prior to DNA-end joining result in <5.1 kb fragments (resected
joints). CH12F3 cells harbor a non-productive rearranged allele resulting in 8.1 kb
amplicon. LR-PCR profiles of Sµ-Sα junctions. IL-2 gene used as a loading control.
Source data are provided as a Source Data file.

rearrangements and *Igk* translocations. Like *p53*−/− *Xrcc4*−/−, all inde-
pendent released/cycling *p53*^−/− *Xrcc4*^−/− *Rhno1*^−/− pro-B clones robustly
recombine the *Igk* locus resulting in similar PCR amplification profiles
obtained for *p53*^−/− *Xrcc4*^−/−-double deficient cells (Fig. 6b). This
demonstrates that Pol θ-mediated repair is active in both XRCC4- and
XRCC4/RHINO-deficient p53-null *v-Abl* pro-B (Fig. 6b). Consistently,
*p53*^−/− *Xrcc4*^−/− *Rhno1*^−/− pro-B cells also contain high levels of *Igk*
translocations (26.42 ± 4.96% of total metaphases), like *p53*^−/− *Xrcc4*^−/−
cells (22.23 ± 7.89 % of total metaphases) and in sharp contrast to *p53*^−/−
*Xrcc4*^−/− *Polq*^−/− cells (0.44% of total metaphase) (Fig. 6c, Supplementary
Data 7), demonstrating that TMEJ of RAG-induced DSBs also occurs
independently of RHINO.

PLK1 regulates TMEJ by phosphorylating Pol θ, which is essential
for its recruitment to mitotic DSBs[8]. As a result, PLK1 inhibition blocks
TMEJ during mitosis. To test whether PLK1 is required for TMEJ at G1-
induced breaks, we exposed ABLki-treated pro-B cells (*p53*^−/−, *p53*^−/−
*Xrcc4*^−/− and *p53*^−/− *Xrcc4*^−/− *Polq*^−/−) with either Polθi or PLK1i

(Supplementary Fig. 9b) and monitored *Igk* repair and instability. Polθi
treatment mimicked the above-described phenotype of Pol θ knock-
out cells (Supplementary Fig. 9, Fig. 6h). By contrast, in NHEJ-deficient
cells, PLK1i (50 nM) still allowed robust *Igk* V_{10-95}-J_4 rearrangement
(Supplementary Fig. 9c) and high *Igk* instability (49.9 ± 9.5%), with
frequent translocations (21.9 ± 5.2% of total metaphases), like DMSO-
treated controls (57.9 ± 0.6 % aberrant metaphases, 32.1 ± 0,9% with
translocations) (Fig. 6h, Supplementary Data 7). This indicates that
TMEJ repair of G1-induced DSBs does not depend on PLK1, supporting
the idea that these events occur outside of mitosis.

We previously showed that RAG-induced DSBs generated in G0/
G1-arrested pro-B cells cause synthetic lethality between XRCC4 and
Pol θ during cell cycle entry[16]. Thus, we analyzed cell cycle dynamics via
EdU/PI staining and cell counting in *p53*^−/− *Xrcc4*^−/− *Rhno1*^−/− pro-B cells
as compared with *p53*^−/− *Xrcc4*^−/− and *p53*^−/− *Xrcc4*^−/− *Polq*^−/−. In cycling
conditions (NT) *p53*^−/− *Xrcc4*^−/− *Polq*^−/− pro-B cell lines proliferate slower
translating into fewer S-phase cells than in *p53*^−/− *Xrcc4*^−/− and *p53*^−/−

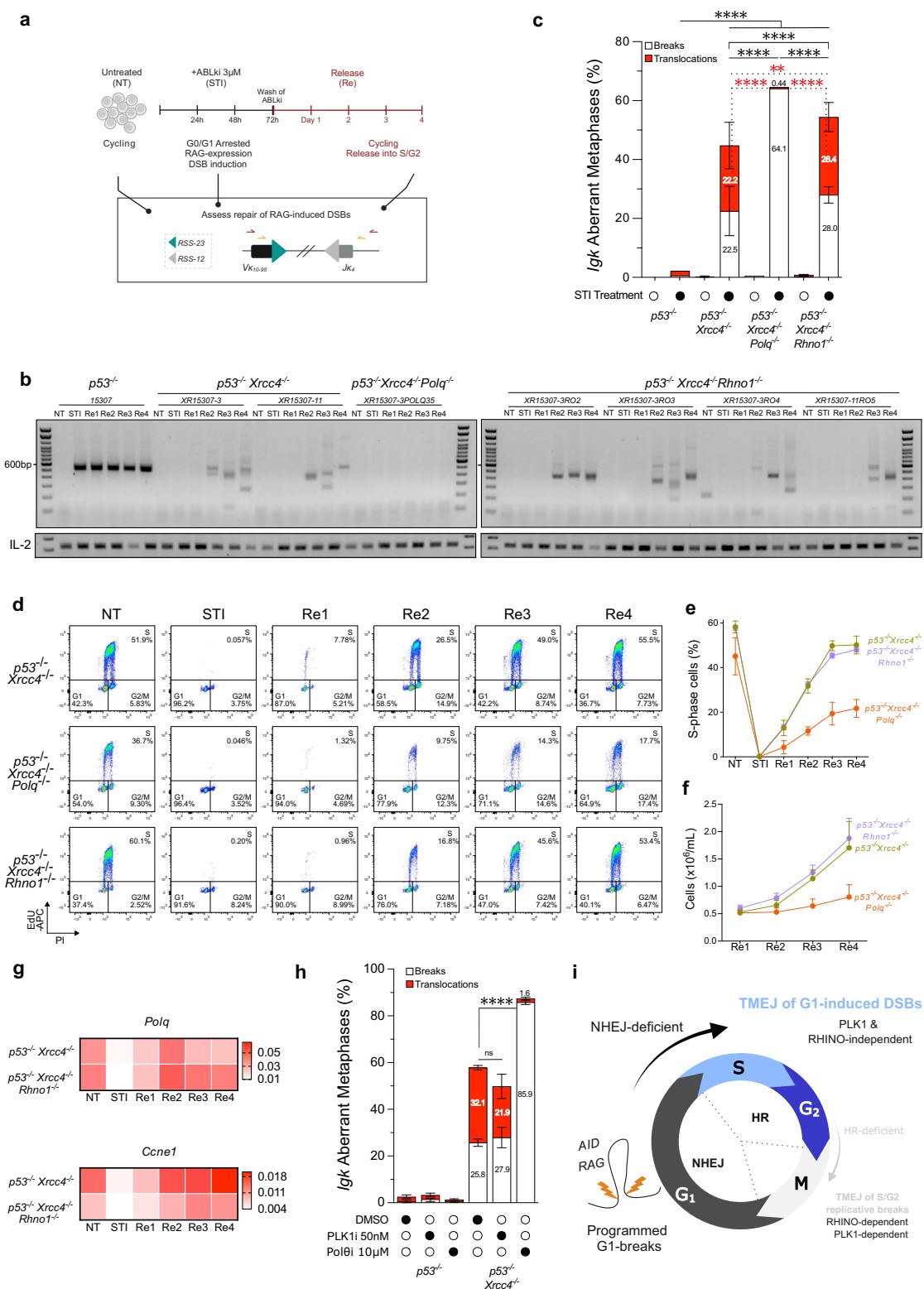

*Xrcc4⁻/⁻ Rhno1⁻/⁻* (Fig. 6d). Upon induction of RAG-generated DSBs in G0/G1 and release into the cell cycle, p53/XRCC4/Pol θ-deficient pro-B difficultly progress into S/G2 phase and are unable to proliferate thus quickly collapse (Fig. 6d-f). In contrast, p53/XRCC4/RHINO-deficient pro-B cells enter S/G2-phase and actively proliferate/divide displaying an undistinguishable cell cycle entry kinetic from that of *p53⁻/⁻ Xrcc4⁻/⁻*

cells (Fig. 6d-f). We also found that G0/G1-arrested p53/XRCC4-deficient cells treated with Polθi show reduced ability to release into S/G2 phase, 20% of cells in S phase by day 2 after G1 release, as compared to 33% DMSO-treatment (Supplementary Fig. 9d, e). Additionally, treatment significantly impaired proliferation relative to DMSO-treated controls, specifically in *p53⁻/⁻ Xrcc4⁻/⁻* cells (Supplementary Fig. 9f). Our

**Fig. 6 | RAG-induced DSBs exiting G1 are repaired in G1/S via TMEJ, independently of RHINO and PLK1. a** Schematic of RAG expression in G1-arrested *v-Alb* pro-B cells with ABLki treatment and release. Analyses from independent cell lines: *p53*[-/-] (15307), *p53*[-/-] *Xrcc4*[-/-] (XR15307-3, XR15307-11), *p53*[-/-] *Xrcc4*[-/-] *Polq*[-/-] (XR15307-3-POLQ35) and *p53*[-/-] *Xrcc4*[-/-] *Rhno1*[-/-] (XR15307-3RO2, XR15307-3RO3, XR15307-3RO4, XR15307-11RO5). **b** Nested PCR of *Igk* V$_{10-95}$-J$_4$ join in untreated (NT), G1 blocked (STI) and released/cycling (Re) pro-B cells. IL-2 as loading control. **c** *Igk* aberrant metaphase frequencies in NT and Re3 cells. Mean percentages (bars) of metaphases with translocations (red) and chromosomes breaks (white) ± SEM from 1-5 independent clones. Total metaphases analyzed in NT and Re3, respectively: *p53*[-/-] (n = 116, n = 504), *p53*[-/-] *Xrcc4*[-/-] (n = 400, n = 986), *p53*[-/-] *Xrcc4*[-/-] *Polq*[-/-] (n = 201, n = 454), and *p53*[-/-] *Xrcc4*[-/-] *Rhno1*[-/-] (n = 1106, n = 1661). Two-sided Fisher's Exact tests compared normal vs. aberrant metaphases, break vs. others and, translocations vs. others (dashed lines) for each genotype. *p* values are indicated in Supplementary Data 7. **d** Representative cell cycle FACS profiles from the block/release assay of pro-B cells. **e** Quantification of S-phase cells throughout the block/ release experiments. Means ± SEM from two independent experiments, *p53*[-/-] *Xrcc4*[-/-] (n = 4), *p53*[-/-] *Xrcc4*[-/-] *Polq*[-/-] (n = 2), and *p53*[-/-] *Xrcc4*[-/-] *Rhno1*[-/-] (n = 10). Two-tailed Mann-Whitney test on AUC (**** *p* < 0.0001). **f** Cell counts of release kinetic, post wash-off of ABLki. Dots represent mean ± SEM from two independent experiments, *p53*[-/-] *Xrcc4*[-/-] (n = 3), *p53*[-/-] *Xrcc4*[-/-] *Polq*[-/-] (n = 2), and *p53*[-/-] *Xrcc4*[-/-] *Rhno1*[-/-] (n = 8). Two-tailed Mann-Whitney test on AUC (* *p* = 0.039). **g** Heat maps of *Polq* and *Ccne1* relative expression during block/release, using RT-qPCR. Data normalized to ACT1and HPRT using 2-(ΔΔCT) method. **h** *Igk* aberrant metaphases frequency in Re3 cells exposure to PLK1i (50 nM) or Polθi (10μM). Mean percentages (bars) of metaphases with translocations (red) and chromosomes breaks (white) ± SEM from 2-3 independent clones. Two-sided Fisher's Exact test compared break vs. others for each genotype, *p* values are indicated in Supplementary Data 7. Total metaphases analyzed: DMSO *p53*[-/-] (n = 543), DMSO *p53*[-/-] *Xrcc4*[-/-] (n = 339), PLK1i *p53*[-/-] (n = 338), PLK1i *p53*[-/-] *Xrcc4*[-/-] (n = 388), Polθi *p53*[-/-] (n = 467), Polθi *p53*[-/-] *Xrcc4*[-/-] (n = 331). **i** Working model for repair of G1-induced DSBs via TMEJ. Source data are provided as a Source Data file.

data demonstrate that the synthetic lethality between NHEJ and Pol θ does not rely on a RHINO-dependent pathway, unlike previous work regarding the synthetic lethality of HR and Pol θ[13]. Notably, *v-Abl* pro-B cells are highly sensitive to PLK1i treatment in both cycling and block/release setting (Supplementary Fig. 9a, f), consistent with its roles during mitosis but also in regulating cell cycle progression and responding to replication stress during S-phase[45].

We then sought out to dissect precisely in which cell cycle phase TMEJ acts on unrepaired G1-induced breaks. Virtually no Pol θ mRNA is detectable in G0/G1-blocked pro-B cells and Pol θ is expressed during cell cycle entry[16] (Fig. 6g). We observe that a first wave, roughly 30%, of synchronized G0/G1-arrested pro-B cells slowly exit G1-phase entering S-phase by day 2 post release (Re2) (Fig. 6d, e). This coincides with the amplification of recombination products in *p53*[-/-] *Xrcc4*[-/-] and *p53*[-/-] *Xrcc4*[-/-] *Rhno1*[-/-] cells (Fig. 6b), and increased expression of Pol θ and Ccne1 (Fig. 6g), the transcriptionally regulated Cyclin E involved in regulation of G1-S phase transition. We performed block/release experiments using two independent *p53*[-/-] *Xrcc4*[-/-] clones and a *p53*[-/-] pro-B cell line as a control. Because released cells only doubled in number at day 3 post ABLki wash off (Fig. 6f), we sorted cells into G1, S, and G2/M populations at day 2 (Re2) or day 3 (Re3) post-G0/G1 release (Supplementary Fig. 10a). Genomic DNA was extracted from non-treated cycling cells (NT), G0/G1-arrested cells (STI), bulk (non-sorted) released cells, and each sorted population. We then performed nested PCR for *Igk* V$_{10-95}$ J$_4$ rearrangements to identify when joins first become detectable, indicating the timing of TMEJ-mediated repair. As expected, in the presence of NHEJ, RAG-induced breaks were rapidly repaired in G0/G1, as indicated by the strong amplification signal in STI samples. This signal persisted across all cell cycle phases following release (Supplementary Fig. 10b). In contrast, NHEJ-deficient cells showed no evidence of *Igk* rearrangement in STI instead, signal accumulated at Re2 and Re3 post-release. When analyzing sorted cell cycle fractions, we observed that a small number of TMEJ-dependent *Igk* joints emerged as early as G1-phase. However, most rearrangements were detected in S-phase fractions (Supplementary Fig. 10b). The few joints amplified in the G2/M populations, likely reflect repair events that occurred in G1- or S-phase in cells that had since progressed to G2/M (Supplementary Fig. 10b). *Igk* V$_{10-95}$-J$_4$ amplicons were TOPO-cloned and sequenced to analyze repair junctions. Consistent with a previous study[16], joins from p53/XRCC4 double-deficient pro-B cells present resection and MH, independently of cell cycle fraction (G1-, S-, G2/M-phase), (Supplementary Fig. 10c, d). Thus, our data indicate that G1-induced breaks were quickly processed and repaired using MH, indicating that TMEJ is active in G1/S-phase. Overall, these data reveal that in the absence of NHEJ, physiological G1-induced breaks are repaired through TMEJ in a RHINO- and PLK1- independent manner, in G1/S-phase prior to entry into mitosis.

## Discussion

ATM-dependent DDR is activated by AID-induced breaks, ensuring efficient CSR through NHEJ[27]. Disrupting NHEJ or its regulation—via defects in DSB sensing, chromatin conformation, end synapsis or protection—reduces switching efficiency while increasing MH usage, resection and inversional recombination[22-24,29,30,39,41,46]. These impairments compromise productive CSR, enhance reliance on alt-EJ pathways, and elevate unproductive CSR, where IgM is lost without generating a functional isotype. We demonstrate that unproductive class switching is fully dependent on Pol θ, repairing resected and inverted joints, while it does not impact productive class switching levels, in NHEJ-deficient and SHLD-unprotected naïve mature B cells. Additionally, we show that B cell programmed DSBs (AID/RAG) repaired by TMEJ occurs in G1/S-phase and does not rely on RHINO or PLK1, indicating that G1-induced breaks are not carried throughout multiple cell cycle phases until mitosis. We highlight the presence of two distinct types of TMEJ, RHINO/PLK1-independent (occurring in G1/S) and RHINO/PLK1-dependent (occurring in M), both responsible for repair in contexts where DSBs travel into the next cell phases (Fig. 6i).

AID-induced breaks and the molecular mechanisms supporting CSR are essential for adaptive immunity, enabling B cells to modify antibody effector functions. While naïve mature B cells primarily use NHEJ to repair AID-DSBs, they can also utilize alt-EJ. In the absence of NHEJ, stimulated B cells robustly rearrange the *Igh* locus, often leaving behind alt-EJ repair signatures, such as ≥4 bp MH stretches. We find that Pol θ is dispensable for productive CSR in XRCC4- and SHLD1-deficient cells. LIG1[47], LIG3[nuc47], XRCC1[48,49], and CtIP[38] are also not required for CSR in NHEJ-deficient B cells. We determine, however, that Pol θ generates the majority of unproductive repair events in NHEJ-deficient and SHLD-unprotected switching B cells, with *Xrcc4*[-/-] *Polq*[-/-] and *Shld1*[-/-] *Polq*[-/-] B cells lacking highly resected and/or inverted joints. Interestingly, this is also the case in B cells deficient for both XRCC4 and CtIP[38], suggesting that CtIP is involved in TMEJ of AID-DSBs. Our findings show that *Xrcc4*[-/-] *Polq*[-/-] and *Shld1*[-/-] *Polq*[-/-] B cells exhibit junctional repair signatures identical to WT, indicating that ≥4 bp MH joints in NHEJ-deficient or SHLD-unprotected settings arise from TMEJ. While we cannot rule out the possibility that in XRCC4- and SHLD1-deficient cells, some productive Sμ-Sy1 joints may be processed through TMEJ, the unchanged CSR levels upon Pol θ abolition indicate that it does not play a major role in productive isotype switching and thus that another repair mechanism exists. Our high-throughput junctional signature analysis further reveals that Pol θ-independent repair predominantly relies on 0–2 bp MH, resembling NHEJ-like repair. Interestingly, in LIG4-deficient B cells, loss of XPF-ERCC1 significantly diminishes productive CSR without significantly impacting MH-driven resected and/or inverted joints[50], suggesting that this complex is involved in the Pol θ-independent repair branch.

Additionally, RAD52 loss significantly reduces CSR efficiency in KU80-deficient B cells[51] and HMCES loss significantly diminishes CSR efficiency in LIG4-deficient B cells[52], though class-switched junctions remain to be analysed in these cells.

While AID-induced DNA breaks are predominantly generated in the G1-phase, CSR occurs in actively proliferating B cells, with AID breaks persisting beyond G1 and thus encountering additional DNA end-processing factors in S and G2 phases[26,27,53–56]. This broadens the spectrum of available repair pathways, increasing the likelihood of resection/homology-directed repair. The presence of RPA at AID-induced ssDNA regions during S-phase further suggests that these breaks are not exclusively repaired by NHEJ but remain accessible to alternative repair mechanisms[54]. Our findings support this, as we observe that the subset of AID breaks which undergo resection is channelled into TMEJ. A key observation is that Pol θ is absent in G0/G1-arrested cells but its expression rapidly increases upon S-phase entry ([16] and this study), indicating that TMEJ is not an immediate G1 repair mechanism. Pol θ has been shown to displace RPA from ssDNA[4], reinforcing the idea that TMEJ occurs predominantly in S or later phases when RPA-coated intermediates are present. Furthermore, Pol θ interacts with several HR and replication stress factors, suggesting that it is active in S-phase[8]. We previously showed that RAG-induced breaks persisting beyond G1 become highly reliant on Pol θ for repair, frequently resulting in translocations[16]. Using the same pro-B cell model, we addressed the timing of G1-break repair via TMEJ and showed repair of RAG-breaks via Pol θ during the G1-to-S-phase transition. This underscores the rapid engagement of TMEJ upon S-phase entry and further highlights its role in maintaining genomic stability when unresolved G1 breaks progress through the cell cycle.

To gain deeper insight into the regulation of G1/S-phase TMEJ, we investigated whether it depends on RHINO or PLK1, two factors critical for Pol θ recruitment during mitotic repair[8,13]. Both RHINO and PLK1 expression are tightly regulated throughout the cell cycle, with their highest levels observed in M-phase, though they begin to accumulate gradually from S/G2[13,45]. Given their established role in mitotic repair, we sought to determine whether RHINO or PLK1 also facilitate the repair of AID- and RAG-induced G1 breaks that persist into S-phase. Through genetic analyses, we demonstrate that RHINO is dispensable for the repair of these breaks, reinforcing the idea that AID- and RAG-induced lesions do not persist until mitosis and are resolved earlier in the cell cycle (G1/S). Additionally, using PLK1 inhibition, we also demonstrated that this repair is independent of PLK1. Thus, our findings suggest that Pol θ recruitment in G1/S-phase operates through a distinct, RHINO- and PLK1-independent mechanism, highlighting fundamental differences between G1/S-phase and mitotic TMEJ. Our findings also imply that G1/S-phase TMEJ of G1-breaks is mechanistically distinct mitotic TMEJ. Unlike in the context of S/G2 replicative breaks in BRCA-deficient cells, TMEJ of G1-induced breaks in NHEJ-deficient cells does not appear to be blocked by RAD52, nor delayed until onset of mitosis[8,13,57]. Moreover, during mitosis, EXO1 degradation by Cyclin-F is essential to permit TMEJ[58]. In S-phase, where the long-range resection exonuclease EXO1 is present, TMEJ does not appear to be inhibited. These results underscore a unique regulatory framework for Pol θ-mediated repair in G1/S-phase, distinguishing it from the mechanisms governing its function in mitotic DNA repair.

RHINO's involvement in MH-mediated end-joining was identified through a targeted genetic screen designed to dissect MH-mediated repair by using cells deficient in NHEJ (*Lig4$^{-/-}$*), HR (*Brca2$^{-/-}$*), and p53 (*Trp53$^{-/-}$*)[13]. This screen identified RHINO as an essential factor for mitotic TMEJ, as its loss in BRCA2-null cells recapitulates the synthetic lethality observed in BRCA2/Pol θ-deficient settings. However, unlike HR-deficient cells, NHEJ-deficient cells do not exhibit sensitivity to RHINO loss; *Xrcc4$^{-/-}$ Rhno1$^{-/-}$* B cells remain viable without signs of severe cellular distress, although they do show a slight increase in baseline genetic instability (as observed in metaphase spreads). Moreover, *p53$^{-/-}$ Xrcc4$^{-/-}$ Rhno1$^{-/-}$* pro- B cells, subjected to RAG-induced G1-phase breaks, resume the cell cycle like *p53$^{-/-}$ Xrcc4$^{-/-}$* cells and unlike *p53$^{-/-}$ Xrcc4$^{-/-}$ Polq$^{-/-}$* cells, which collapse upon entry into the cell cycle. This suggests that RHINO's identification in the screen was primarily due to its synthetic lethality with HR rather than NHEJ.

Overall, we reveal that in the absence of NHEJ, productive isotype class switching occurs independently of Pol θ while unproductive recombination events are Pol θ-dependent, both being important for preserving *Igh* locus integrity. Furthermore, our study reveals that G1-induced DSBs are repaired by Pol θ in S-phase independently of RHINO and PLK1, distinguishing it from TMEJ of replicative breaks in mitosis. These findings shed light on the mechanistic differences underlying the synthetic lethality of Pol θ with NHEJ versus HR, with potential implications for developing therapeutic strategies that exploit DNA repair vulnerabilities in cancer.

## Methods
### Mice
*CD21-Cre$^{tg}$*, *Xrcc4$^{flox/flox}$*, *Polq$^{tm1Js}$* (B6.Cg-Polqtm1Jcs/J; RRID:IMSR_JAX: 006194) and *Shld1$^{-/-}$* mice were previously described[29,32,33,59]. These mice were crossed to obtain double deficient *Xrcc4$^{-/-}$/Polq$^{-/-}$* and *Shld1$^{-/-}$/Polq$^{-/-}$* individuals. Mice were bred under specific- pathogen-free (SPF) conditions and housed with 12 h light/12 h dark cycles. In all experiments, 6–12-week-old sex- and age-matched mice were used. In absence of the *CD21-Cre$^{tg}$*, *Xrcc4$^{flox/flox}$* or *Xrcc4$^{+/flox}$* were indistinguishable to WT thus, *Xrcc4$^{flox/flox}$* mice were also used as controls. Mice euthanasia was performed by carbon dioxide exposure. All experiments were performed in accordance with the guidelines of the institutional animal care and ethical committee of Institut Pasteur/CETEA (dha210006).

### Lymphocyte development
Lymphocyte development was analyzed in the thymus, bone marrow, and spleen of mice aged 6–12-weeks-old, with matching sex and age. Single-cell suspensions from all tissues were pretreated with Fc-blocking antibody (CD16/CD32 (Fcg III/II Receptor), BD Biosciences 553142, clone 2.4G2, 1:200 dilution) before cell surface staining. All antibody staining's were performed in phosphate-buffered saline (PBS). Flow cytometry was acquired on a FACS Fortessa (BD Bioscience), and data analyzed using FlowJo™ v10.8.0 (TreeStar) according to the gating in Supplementary Fig. 1.

B lineage cell populations were identified based on the expression of the following markers in bone marrow samples: immature B cells (B220$^{low}$ IgM$^+$), recirculating B cells (B220$^{high}$ IgM$^+$), pro-B (B220$^{low}$ CD43$^+$ IgM$^-$) and pre-B (B220$^{low}$ CD43$^-$ IgM$^-$). T lineage cell populations from the thymus were identified based on the following expression profiles: double-negative (DN) cells (CD4$^-$CD8$^-$), DN1 (CD4$^-$CD8$^-$CD44$^+$CD25$^-$), DN2 (CD4$^-$CD8$^-$CD44$^+$CD25$^+$), DN3 (CD4$^-$CD8$^-$CD44$^-$CD25$^+$), DN4 (CD4$^-$CD8$^-$CD44$^-$CD25$^-$), double-positive (DP) cells(CD4$^+$CD8$^+$) and single-positive cells (CD4$^+$CD8$^-$ and CD4$^-$CD8$^+$). Lymphocytes from the spleen were identified based on the expression of the following markers: total B cells (CD19$^+$IgM$^+$) and T cells (CD3$^+$TCRβ$^+$). The succeeding antibodies were used for cell surface staining: anti-B220-AF488 (BD Pharmingen™ 557669, clone RA3–6B2, 1:200 dilution), anti- CD43-PE (BD Pharmingen™ 553271, clone S7, 1:200 dilution), anti-CD19-V450 (BD Horizon™ 560375, clone 1D3, 1:200 dilution), anti-IgM-PECy7 (BD Pharmingen™ 552867, clone R6–60.2, 1:200 dilution), anti-CD4-PE (BD Pharmingen™ 553048, clone RM4–5, 1:200 dilution), anti-CD8a-AF488 (BD Pharmingen™ 557668, clone 53–6.7, 1:200 dilution), anti- CD3e-APC (BD Pharmingen™ 553066, clone 145-2C11, 1:200 dilution), anti-CD44-V450 (BD Horizon™ 560451, clone IM7, 1:200 dilution), anti-CD25-PECy7 (BD Pharmingen™ 552880, clone PC61, 1:200 dilution), and anti-TCRβ-APC-eF780

(eBiosciences 47-5961-82, clone H57–597, 1:200 dilution). For full gating strategy, refer to[29].

## Ex vivo CSR assay

Splenic B cells were purified from mice using magnetic anti-CD19 beads according to the manufacturer's instructions (Miltenyi Biotec 130-121-301). Cells were cultured at 37 °C, 5% $CO_2$, in complete medium containing RPMI 1640 supplemented with 12% FBS, penicillin (100 U/ml)/streptomycin (100 µg/ml) and 50 µM 2-mercaptoethanol. CSR assays were conducted at $0.6 \times 10^6$ cells/mL over 4-5 days using the following cytokine combinations; IgG1 with anti-IgD dextran (3 ng/ml, Fina Biosolutions), IL-4 (10 ng/ml, Miltenyi) and LPS (25 µg/ml; Sigma-Aldrich), IgG2b and IgG3 with anti-IgD dextran (3 ng/ml) and LPS (25 µg/ml) and IgA with anti-IgD dextran (3 ng/ml), IL-5 (5 ng/ml; R&D Systems), TGF-β (3 ng/ml; R&D Systems) and retinoic acid (RA; 0.3 ng/ml; Sigma-Aldrich). Class switching was evaluated by flow cytometry using anti-CD19-V450 (BD Horizon™ 560375, clone 1D3, 1:200 dilution), anti-CD19-APC (BD Pharmingen™ 550992, clone 1D3, 1:200 dilution), anti-IgG1-APC (BD Pharmingen™ 550874, clone X56, 1:200 dilution), anti-IgG2b-PE (Biolegend 406708, clone RMG2b-1, 1:200 dilution), anti-IgG3-FITC (BD Pharmingen™ 553403, clone R40-82, 1:200 dilution), anti-IgM-PE-Cy7 (BD Pharmingen™ 552867, clone R6–60.2, 1:200 dilution) and anti-IgA-PE (eBioscience™, clone mA-6E1, 1/200 dilution) antibodies. Flow cytometry was acquired on a FACS Fortessa (BD Bioscience), and data analyzed using FlowJo™ v10.8.0 (TreeStar).

## Cell Proliferation (CTV)

Proliferation analysis via in vitro cell labeling was performed on purified splenic CD19$^+$ cells that were stained at $5 \times 10^6$ cells/ml in PBS with 5 µM Cell Trace™ violet (CTV) for 8 min at room temperature (Thermo Scientific 10220455), following the manufacturer's protocol. Two additional washes with complete medium were added prior to incubating CTV-stained B cells for 1 h at 37 °C. They were then stimulated with LPS, IL-4, and anti-IgD dextran as described above. FACS data was acquired on a Fortessa analyzer (BD Biosciences) and analyzed and visualized by Flowjo software v10.4.2 (TreeStar).

## DNA-FISH probes

BAC probes for the *Igh* locus, BAC RP24-134G24 (*IghC*) and BAC RP24-386J17 (*IghV*), and for the *Igk* locus, BAC RP23-341D5 (*IgkC*) and BAC RP24-243E11 (*IgkV*), were directly labeled by nick-translation with ChromaTide Alexa Fluor 488-5–dUTP or ChromaTide Alexa Fluor 594-5–dUTP (Molecular Probes). Labeled BAC probes were purified using Illustra MicroSpin G-50 Columns (GE Healthcare). A total of 800 ng of each BAC probes were pooled and precipitated with mouse *Cot1* DNA and Salmon Sperm DNA (Life Technologies). Probes were then re-suspended in hybridization buffer (10% dextran sulfate, 5x Denharts solution, 50% formamide), denatured for 5 min at 95 °C and pre-annealed for 1 h at 37 °C. XCyting Mouse Chromosome 12 (Orange) or XCyting Mouse Chromosome 6 (Orange) paint from MetaSystems was denatured for 5 min at 95 °C and mixed with BAC probes just before hybridization on metaphases spreads.

## DNA-FISH on metaphases spreads

Metaphases were prepared using standard procedures. Briefly, cells were incubated with colcemid (0.03 µg/ml, Life technologies, KaryoMAX Colcemid Solution) for 4 h at 37 °C. Then, cells were collected and incubated in 0.075 M KCl for 20 min at 37 °C, fixed in fixative solution (75% methanol/25% acetic acid) and washed three times in the fixative solution. Cell suspension was dropped onto ice-cold humid slides and air-dried for further processing.

DNA-FISH was performed on metaphase spreads by treating slides with RNase A (0.025 µg/µl in 0.5× SSC) for 30 min, dehydrating in 70, 90, and 100% ethanol (3 min/bath), denaturing in 70% formamide/2x SSC (3 min, 77 °C), dehydrating in ice-cold ethanol bath series (70, 90, 100% ethanol – 3 min/bath), and hybridizing with probes (overnight, 37 °C humid chamber). Slides were washed in 50% formamide/0.5x SSC at 37 °C (3 × 5 min each) and in 0.5x SSC at 37 °C (1 × 10 min, 1 × 20 min). Subsequently, dry slides were mounted in ProLong Gold antifade reagent with DAPI (Invitrogen P36931). Metaphases were imaged using a ZEISS AxioImager.Z2 microscope and the Metafer automated capture system (MetaSystems). Analysis of metaphases was performed manually on Metafer.

## LR-PCR and PacBio sequencing

Long-range PCR were conducted using Platinum SuperFi II polymerase MasterMix (Invitrogen 16445389). A total of 100 ng of gDNA was amplified using EµF and Cγ1R2 or CαR primers. Cycling used: 1x (98 °C 30 s); 30x (98 °C 10 s, 57 °C 10 s, 68 °C 8 min); 1× (68 °C 5 min). Loading control, IL-2 gene was amplified using IMR42 and IMR43 primers and the following cycles: 1× (95 °C 5 min); 28× (95 °C 30 s, 56 °C 30 s, 72 °C 30 s); 1× (72 °C 5 min). Refer to Supplementary Data 9 for primer sequences.

For each sample, bulk PCR amplicons were purified, and sequencing libraries prepared, prior to sequencing using the long-read PacBio Revio technology by Novogene. Sequences were demultiplexed and consensus circular reads (CCS) were constructed using the SMRTLINK software from PacBio (v9) (smrtlink: https://www.pacb.com/support/software-downloads/). After conversion of CCS reads into FastQ files, they were mapped onto the Mus musculus genome (reference GRCm38) using minimap2 v2.26-r1175 including PacBio and splicing options. Reads which did not align to both 5' and 3' sections of the region of interest (IgM-to IgG1: chromosome 12, 113330523 – 11333884; IgM-to-IgA: chromosome 12, 113255485 – 113431785) were filtered out. Subsequent bioinformatics analysis of the aligned reads was performed with SAMtools and R for coverages and read lengths, while junction analyses were conducted using Python and R scripting. Briefly, for each read, the breakpoint spanning from 5' (Sα/Sy1 regions ± 5 kb) to 3' end (Sµ regions ± 5 kb) was identified, their genomic location and CIGAR information was extracted. To counteract the potential PCR bias introduced by the LR-PCR, only unique break-points with a ≥10x sequencing depth were used for downstream analysis of repair location and signature. Breakpoints located (5' and/or 3' end(s)) outside of either the Sµ or Sα/Sy1 regions were defined as a resected join. Additionally, the rearranged non-productive allele in the CH12F3 cell line[60] was removed from analysis. We modeled the read lengths in a linear mixed model using lmer function from package lme4[61] and estimated the statistical significance of each contrast by the Tukey HSD test using the emmeans package (https://cran.r-project.org/web/packages/emmeans/index.html).

## CSR-HTGTS

HTGTS libraries were generated[29,40,41] (Supplementary Data 9), sequenced using 150 bp paired end Nova- seq (Illumina), and processed[40] using the mm9 genome reference as previously described.

## Culturing of CH12F3 and *v-Abl* pro-B cell lines

CH12F3 and *v-Abl* pro-B cells were cultured in RPMI 1640 supplemented with 12% FBS (Sigma F7524), penicillin (100 U/ml)/streptomycin (100 µg/ml), 50 µM 2-mercaptoethanol, 1x MEM nonessential amino acids, 1 mM sodium pyruvate, and 10 mM HEPES.

We performed sensitivity assays of *v-Abl* pro-B cells using Polθi and PLKi to determine suitable treatment doses that maintain cell viability (Supplementary Fig. 9a), as indicated throughout text and figures we used 5µM/10µM of Polθi (ART558, MedChemExpress, HY-141520) and 50 nM of PLK1i (Volasertib, Euromedex S2235-10mM/1 mL, Bi6727).

## Culture and CRISPR-Cas9 engineered CH12F3 and *v-Abl* pro-B cell lines

Knockout cell clones were generated as previously described[16,44,62]. Plasmid containing two sgRNA-encoding the pCas9-EGFP, specific to each targeted gene, was used to nucleofect CH12F3 and/or *v-Abl* pro-B cells. Electroporated cells were left to recover for 24 h and single cells expressing GFP were sorted in 96-well plates. Clones were then screened by PCR and PCR products were Sanger sequenced to identify CRISPR/Cas9-induced editing at cleavage sites. Several independent knock-out clones were generated (Supplementary Data 8). See Supplementary Data 9 for used sgRNA sequences and genotyping primers. Due to lack of good antibodies against murine XRCC4 and Pol θ, we validated our *Xrcc4*$^{-/-}$ and *Polq*$^{-/-}$ CH12 B cell lines using etoposide and MMS sensitivity assays (Supplementary Fig. 5b). CH12 B cells were exposed to various concentrations of etoposide or MMS during 48 h and viable cell counts were recovered by flow cytometry, using a MACSQuant analyzer (Miltenyi Biotec, Bergisch Gladbach, Germany) and FACS profiles are analyzed using FlowJo software v10.4.2 (TreeStar).

## CSR assay in CH12F3 cells lines

To induce IgM-to-IgA class switching, 50,000 CH12F3 cells/ml were plated in complete RPMI medium supplemented with anti-CD40 antibody (1 μg/ml, Miltenyi), IL-4 (20 ng/ml, Miltenyi) and TGF-β (1 ng/ml, R&D Biotech). Day 4 post cytokine stimulation, class-switching capacity was determined by flow cytometry using IgA-PE (mA-6E1, 1:200) and IgM-PE-Cy7 (R6-60.2, 1:200 dilution) antibodies, and a Fortessa analyzer (BD Biosciences). Data were analyzed by FlowJo software v10.4.2 (TreeStar). GLTs and AID expression levels were assessed across CH12F3 cell lines used (Supplementary Fig. 8).

## Induction of RAG-DSBs

*V-Abl* pro-B cells were treated with 3 μM of the *Abl* kinase inhibitor STI-571 (Novartis) for 72 h, to induce G0/G1 arrest and expression/stabilization of RAG. Cells were washed 3-times with PBS to remove ABLKi, resulting in cell release from G0/G1-phase and S-G2/M entry.

## Igk nested PCR assay, V(D)J recombination product

As previously described, endogenous *Igk* V$_{10-95}$-J$_4$ coding joints were amplified by nested-PCR[16]. A total of 250 ng of genomic DNA was amplified using Jk4-pa and V10-95-pa primers and the following cycles: 1× (95 °C 5 min); 17× (95 °C 30 s, 61 °C 30 s, 72 °C 30 s); 1× (72 °C 5 min). Using Jk4-pb and V10-95-pb primers, 2.5 μl of the first PCR reaction was reamplified using the following cycles: 1× (95 °C 5 min); 30× (95 °C 30 s, 61 °C 30 s, 72 °C 30 s); 1× (72 °C 5 min). Loading control, IL2 gene was amplified using IMR42 and IMR43 primers and the following cycles: 1× (95 °C 5 min); 28× (95 °C 30 s, 56 °C 30 s, 72 °C 30 s); 1× (72 °C 5 min). See Supplementary Data 9 for primers sequences.

PCR products from *Igk* V$_{10-95}$-J$_4$ nested PCR were purified on agarose gel and cloned into TA pCR™4-TOPO® (Invitrogen™, 450030) following manufacturing protocol. Single colonies were isolated and cloned fragments sequenced using SANGER sequencing. Sequences were analyzed with in house R/bash scripting to detect presence of insertions, MH, and resection.

## Cell Cycle

*V-Abl* pro-B cells were incubated with 100 μM Edu (Jena Bioscience CLK-N001-25) for 1 h at 37 °C, washed (PBS 1x) and fixed with 2% Paraformaldehyde at 4 °C for at least 24 h. Cells were stained with 1 mM CuSO4 (Sigma C1297-100G), 2 μM iFluor™ 647 azide (AAT Bioquest 1091) and 100 mM l-Ascorbic acid (Sigma A5960-25G) for 1 h at room temperature and washed with PBS. Then, cells were incubated with PI staining cocktail (1.9 mM sodium citrate tribasic dehydrate (Sigma S4641-500G), 25 μg/ml propidium iodide (ThermoFisher Scientific 440300250), 250 μg/ml RNAse A (Invitrogen 8003089), 0.5 mM Tris-Hcl, 0.75 mM NaCl) overnight at 37 °C. FACS data was acquired on a Fortessa analyzer (BD Biosciences) and analyzed and visualized by Flowjo software v10.4.2 (TreeStar). Cells from each cell cycle phase were sorted using a MoFlo Astrios EQ (Beckman Colter), where 4-20 million cells per gate were isolated for gDNA isolation.

## RT-qPCR, AID and GLTs expression

RNA and cDNA preparation was conducted using standard techniques. Quantitative PCR was performed using SYBR™ Green PCR Master Mix (Applied Biosystems 4309155) in technical duplicates or triplicates and data collected on a QuantStudio 3 Real-Time PCR System. Data was analyzed using the QuantStudio Design & Analysis Software v2.6.0, where the threshold cycle (CT) values were determined. Genes ACT1 and HPRT were used as calibrator and control, respectively. The relative expression of transcripts of interest was determined by comparing the normalized values to WT samples or to HPRT. See Supplementary Data 9 for primer list.

## Reporting summary

Further information on research design is available in the Nature Portfolio Reporting Summary linked to this article.

## Data availability

The HTGTS and PacBio data generated in this study have been deposited in NCBI's Gene Expression Omnibus and Sequence Read Archive (SRA) respectively, under accession numbers GSE306291 (HTGTS) and PRJNA1304899 (PacBio). Source data includes all uncropped gels, and data shown in graphs for all figures and Supplementary Figs. Source data are provided with this paper.

## Code availability

Custom scripts to analyze PacBio data are available at Zenodo (https://doi.org/10.5281/zenodo.17266277)[63].

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

## Acknowledgements

We acknowledge the Institut Pasteur animal facility, Center for Animal Resources and Research (C2RA), specifically Quentin Mille, Gwendal Kerzerho, David Ferreira and Jéromine Rohard Blanco, for mouse breading. We thank Fred Alt for providing the Xrcc4-floxed line. We thank Jean-Pierre de Villartay for providing the CD21-Cre3a line and Klaus Rajewsky and MGC for permission to use the line. We thank the Wellcome Trust Sanger Institute Mouse Genetics Project (Sanger MGP) and its funders for providing the mutant mouse line (Allele: *Shld1em1(IMPC) Wtsi*), and INFRAFRONTIER/EMMA (www.infrafrontier.eu). Funding information may be found at www.sanger.ac.uk/mouseportal and associated primary phenotypic information at www.mousephenotype.org. We also thank the Biomics Platform, C2RT of Institut Pasteur, particularly Chloé Baum and Thomas Cokelaer, regarding PacBio sequencing of LR-PCR. We thank Emeline Perthame of the Bioinformatics and Biostatistics Hub at the Institut Pasteur for her help with the statistical analyses. We thank past and present members of the Deriano lab for their discussion, advice and technical help. Notably, Marie Bedora-Faure for her involvement in breading of CD21-Cre Xrcc4 Polq mice. Some schemes (Fig. 2a Fig. 5a; Fig. 6a, i; Supplementary Fig. 7a; Supplementary Fig. 9b) were created in BioRender. Marton, T. (2025) https://BioRender.com/f4bfpgs. This work was supported by the Institut Pasteur, Institut National de la Santé et de la Recherche Médicale, Ligue Nationale Contre le Cancer (Labellisation 2019–2023 and 2024–2026), Institut Nationale du Cancer (PLBIO19-122, INCa_13852), Worldwide Cancer Research (grant #19-0333), and Agence Nationale de la Recherche (CSR_BreakIR). The Frock lab is supported by the American Cancer Society Research Scholar Grant (RSG-23-1038994-01DMC).

## Author contributions

L.D. and T.M. conceived the study. T.M. performed experiments and analyzed the data. T.M., W.Y. and T.E.C. generated knockout CH12F3 B cell lines. P-H.C. conducted cell sorting of cell cycle populations. Q.H. contributed to DNA-FISH experiments. J.W., T.M. and R.F. performed and analysed LAM-HTGTS experiments. T.M. and A.V. developed the LR-PCR sequencing analysis pipeline. T.M. and L.D. wrote the manuscript with inputs from R.F. All authors read and approved the manuscript.

## Competing interests

The authors declare no competing interest.
