## [Transparent Peer Review file · Nature Communications]

Polymerase theta repairs persistent G1-induced DNA breaks in S-phase during class switch recombination

Corresponding Author: Dr Ludovic Deriano

Version 0:

Reviewer comments:

Reviewer #1

(Remarks to the Author)

This study investigates how DNA breaks are repaired during class switch recombination (CSR) in B cells lacking DNA Double Strand Break repair proteins. It shows that when non-homologous end joining (NHEJ) is disrupted (via XRCC4 or SHLD1 deficiency), polymerase theta (Pol θ) steps in to repair breaks, even though this leads to non-efficient CSR. These alternative repairs occur independently of the RHINO protein and probably during the S/G2 phase, suggesting that in the absence of NHEJ, Pol θ -mediated end joining (TMEJ) handles persistent DNA breaks outside of mitosis.

The work is conducted in a technical sound manner with good attention to detail and solid statistical determinations.

I have only two comments:

1. Although the authors show nicely that the repair of persistent DSBs by NHEJ is not RHINO dependent, it is less clear if they are repaired in S/G2 phase. The presence of long-range resection Exonucleases in S phase (like EXO1), should prevent the action of polQ. It has been shown that polQ can execute DSB repair in mitosis because EXO1 is degraded by cyclin F (<https://www.science.org/doi/10.1126/sciadv.ado0636>). Are the B cells devoid of EXO1 ?
2. The study starts from assessing the role of polQ in CSR, but it diverts on the handling persistent DSBs in S phase by TMEJ. Can the authors reconcile this with more emphasis on the importance of these observation in physiological settings ?

Reviewer #2

(Remarks to the Author)

This study by Marton et al. investigates the mechanisms of double-strand break (DSB) repair during class switch recombination (CSR), a process essential for antibody diversification. To this end, they examine CSR activity in various genetic contexts, specifically in the absence of classical non-homologous end joining (NHEJ) factors (XRCC4 and SHLD1) or the alternative/microhomology-mediated end joining (Alt-EJ) factor, Pol theta (Pol θ), both pathways known to contribute to end-joining during CSR. The study particularly focuses on the role of Alt-EJ, mediated by Pol θ , when NHEJ is impaired. To explore this, they analyze the repair of DSBs induced in the G1 phase, where DSBs are typically resolved by either NHEJ or Alt-EJ pathways. The authors show that, in the absence of NHEJ, Pol theta repairs G1-programmed DSBs that have undergone resection. They further demonstrate that this repair occurs during the S/G2 phase of the cell cycle, rather than during mitosis, as it is entirely independent of Rhino, a recently identified mitotic-specific Pol θ interactor.

The study is well conducted, and the results are clearly presented. The data have the potential to advance the field.

I have only few comments:

The conclusion that Pol θ -mediated repair occurs rapidly upon G1 exit is based on mRNA expression and cell proliferation data, and, in my view, represents an overstatement. Additionally, there is a potential bias in the interpretation, as XRCC4^{-/-}; POLQ^{-/-} cells do not appear to effectively progress through the cell cycle following the induction of RAG-induced DSBs (Fig. 6D). In this context, it might be valuable to use a POLQ inhibitor instead, as this could reduce repair activity while potentially allowing cells to continue progressing through the cell cycle.

Reading the paper, one might get the impression that the authors claim Pol θ functions during mitosis is exclusively a Rhino-dependent. This point should be clarified, as it does not accurately reflect the state of the field; the involvement of Rhino remains preliminary as it is currently supported only by a single publication. Using a PLK1 inhibitor could strengthen the evidence for a mitosis-independent role of Pol θ and help consolidate the findings presented in the study. It would also be valuable to generate Rhino $^{-/-}$; Pol $\theta^{-/-}$ cells to further investigate this relationship.

It is difficult to discern the major novelty of the findings, particularly how they differ from the previous study by the same group, which showed that DSBs generated in G1 in NHEJ-deficient cells are repaired during the following S/G2/M phases by Pol theta (Yu et al., Nat Comm, 2020). This distinction should be addressed and discussed in the manuscript's discussion section.

Reviewer #3

(Remarks to the Author)

The manuscript entitled titled "Polymerase theta-mediated end-joining repairs persistent G1-induced DNA breaks in S/G2" by T. Marton et al. presents important findings that significantly advance our understanding of DNA repair pathway choice across the cell cycle. The authors demonstrate that B cells deficient in both XRCC4 and Pol θ , or SHLD1 and Pol θ , accumulate unresolved IgH double stranded DNA breaks. Surprisingly, these cells retain efficient class switch recombination (CSR), highlighting the role of polymerase theta-mediated end joining (TMEJ).

This study reveals that in the absence of core non-homologous end joining (NHEJ) components, Pol θ mediates aberrant end joining characterized by extensive resection, microhomology usage, and inversion events, independently of the RHINO complex. These observations challenge the prevailing view that TMEJ is confined to mitotic phases, demonstrating instead that it can function during S/G2 to resolve persistent G1-induced DNA breaks. This finding redefines the temporal flexibility of TMEJ and expands our understanding of backup repair pathways. As usual from the Deriano lab, the experimental data has been performed rigorously, presented clearly and including all the appropriate controls and corresponding statistical analysis. Nevertheless, a few items need to be addressed before the manuscript is ready for publication.

Specific comments:

1. Clarification needed regarding switching efficiency: The following statement should be revised for accuracy: 'Furthermore, Pol θ 's dispensability extended beyond IgM-to-IgG1 switching, as cells efficiently recombined toward IgG2b, IgG3, and IgA isotypes in both XRCC4-proficient and XRCC4-deficient B cells (Fig. 1E-G)'. In fact, XRCC4-deficient B cells did not switch efficiently to IgG2b, IgG3, and IgA isotypes compared to XRCC4-proficient (WT) B cells. Please rephrase this sentence to reflect the comparative decrease in switching efficiency in the XRCC4-deficient background.
2. Unclear Smears in Figure 3B: The authors refer to smears below the S-S joints in XRCC4-deficient cells (Figure 3B). However, these smears are not visible in the gel as presented. In contrast, smears are clearly visible in the SHLD1-deficient condition. Please consider repeating the experiment or providing a higher-quality gel image to confirm the results.
3. In Figure 3C, an asterisk appears in the figure without explanation. Please add a corresponding note in the figure legend regarding the asterisk.
4. Missing Panels in Figure S3: The authors reference panels C and D of Supplementary Figure S3 in the Results section, but these panels are absent from the current version of the figure. Please verify and include the missing panels along with an updated legend.
5. The manuscript states that XRCC4 or SHLD1 deletion significantly increases S μ -S μ joints (Fig. 4B, Fig. S4A, B). However, increased S μ -S μ joining is observed only in XRCC4-deficient cells (Fig. S4A). SHLD1-deficient cells show a reduction compared to WT (Fig. S4A). Please revise this section to accurately describe the data.
6. To establish that TMEJ is responsible for repairing RAG-induced double-strand breaks (DSBs) specifically in S/G2-phase cells, it is essential to show that TMEJ-type junctions are more prevalent in S/G2-phase cells. This can be achieved by combining cell cycle analysis with molecular characterization of DNA repair junctions. Additionally, it is important to demonstrate that features of Pol θ -mediated repair, such as microhomology, resection, and insertions, are predominantly found in S/G2-phase cells. A cell cycle analysis using FACS sorting to isolate S/G2-phase cells, followed by LR-PCR to amplify and sequence RAG-induced junctions could be performed. Compare the frequency of TMEJ features (microhomology, resection, and insertions) in S/G2-phase cells versus mitotic cells to demonstrate preferential repair by Pol θ in S/G2.
7. Genotyping Band Size Error in Supplementary Figure S1A; the size of the WT band for XRCC4 genotyping is incorrectly indicated. Please correct the band size.
8. Mismatch Between Figure and Text (Figure 2B): In the results part, specifically the paragraph about TMEJ joints a subset of AID-induced DNA breaks during CSR, the authors mention in the text the mean numbers of aberrant metaphases and refer to Figure 2B and supplementary Table S1. However, the authors represented the number of translocations and breaks in Figure 2B. I suggest adding legend for breaks and translocations bars on the figure itself as Figure 5C and refer only to

table S1 in the text for the number of aberrant metaphases.

9. Inconsistent Use of Units (bp vs. Kb): The authors use both “bp” and “Kb” when referring to fragment sizes in the same section. For clarity and consistency, please standardize the unit usage throughout the manuscript.

10. The authors refer to Figure 4B and 4D when describing the percentage of resected joints in WT and Polq^{-/-} IgG1-stimulated cells. However, this data is shown in Figure 4C and 4D. Please correct the citation accordingly.

11. The authors described the % of S_μ-S_{γ1} direct and possess MH>4bp joints by citing only Figure 4E. Please also reference Figure 4F, which presents the data for S_μ-S_{γ1} joins with MH >4 bp. The sentence, “GLTs and AID expression levels were assessed across CH12F3 cell lines used (Fig. S7),” appears in the CSR assay in CH12F3 cells lines section. It would be more appropriate to include this sentence in the paragraph related to RT-qPCR for AID and GLT expression.

12. Please correct the genotype annotations in Supplementary Tables S1, S2, S3, S4 and S5 to maintain consistency with Tables S6 and subsequent tables. For example, “polq” should be written as “polq^{-/-}” and etc...

Version 1:

Reviewer comments:

Reviewer #1

(Remarks to the Author)

The authors have addressed the comments and the manuscript is now suitable for publication.

Reviewer #2

(Remarks to the Author)

The manuscript significantly improved during the revision process, particularly with the inclusion of novel data and controls. Altogether this manuscript provides strong evidence for a role of POLQ in repairing programmed G1 breaks during the subsequent S phase rather than in mitosis.

I have no further comments and am pleased to strongly recommend the revised manuscript for publication.

Reviewer #3

(Remarks to the Author)

The authors have addressed all my concerns and modified the manuscript accordingly.

I believe the manuscript is ready for publication.

August 25, 2025

Response to the reviewers' comments

We thank the referees for their careful review of the manuscript, their positive comments on our work and their insightful suggestions for improving our article. What follows are point-by-point responses (in blue) to the reviewers' comments (in black).

Reviewer #1 (Remarks to the Author):

This study investigates how DNA breaks are repaired during class switch recombination (CSR) in B cells lacking DNA Double Strand Break repair proteins. It shows that when non-homologous end joining (NHEJ) is disrupted (via XRCC4 or SHLD1 deficiency), polymerase theta (Pol θ) steps in to repair breaks, even though this leads to non-efficient CSR. These alternative repairs occur independently of the RHINO protein and probably during the S/G2 phase, suggesting that in the absence of NHEJ, Pol θ -mediated end joining (TMEJ) handles persistent DNA breaks outside of mitosis.

The work is conducted in a technical sound manner with good attention to detail and solid statistical determinations.

I have only two comments:

1. Although the authors show nicely that the repair of persistent DSBs by NHEJ is not RHINO dependent, it is less clear if they are repaired in S/G2 phase. The presence of long-range resection Exonucleases in S phase (like EXO1), should prevent the action of polQ. It has been shown that polQ can execute DSB repair in mitosis because EXO1 is degraded by cyclin F (<https://www.science.org/doi/10.1126/sciadv.ado0636>). Are the B cells devoid of EXO1?

..it is less clear if they are repaired in S/G2 phase.

We thank the reviewer for his/her comment – a point also raised by referee #2 and #3.

We now provide direct evidence that repair of persistent G1-induced breaks occurs in S-phase, starting during the G1-to-S transition. As suggested by referee #3, we used *p53^{-/-} Xrcc4^{-/-}* pro-B cell clones to perform G0/G1-block (*i.e.* RAG-induced DSBs) and release experiments, followed by FACS sorting of G1, S, and G2/M populations, this to directly assess TMEJ during pro-B cell entry into G1-S (Supplementary Fig. 10). Cells were collected at day 2 and day 3 post-ABLki washoff, corresponding to the first wave of cells exiting G1 into S-phase (Fig. 6c), and cell fractions were marked using EdU/PI before sorting. To track TMEJ, we measured *Igk* V₁₀₋₉₅ J₄ recombination and detected joins predominantly in S-phase, with a few arising as early as during the G1-to-S phase transition (Supplementary Fig. 10b). We cloned and sequenced recovered recombination products, revealing the presence of resection and microhomology usage, which is consistent with TMEJ (Supplementary Fig. 10c-d). Together, these results demonstrate that TMEJ is active during S-phase, and thus that repair of G1-induced breaks is not delayed until mitosis.

The presence of long-range resection Exonucleases in S phase (like EXO1), should prevent the action of polQ.

Indeed, B cells should express EXO1, and in fact, it has been reported that EXO1 is essential for CSR (Sun X et al, Front. Cell Dev. Biol., 26 November 2021, PMID: 34926456, DOI: 10.3389/fcell.2021.767624). Prior work has demonstrated that CCFN-mediated degradation of EXO1 in G2/M allows TMEJ to occur predominantly in mitosis (Yang H et al, Sci Adv, 2024 Aug 9, PMID: 39121215, DOI: 10.1126/sciadv.ado0636). In that study, a non-degradable EXO1 mutant impaired microhomology mediated end-joining and favored SSA, consistent with the idea that extensive EXO1-driven resection counteracts Pol θ -mediated repair. In our cellular systems, however, we find that TMEJ is active in S-phase, indicating that TMEJ can occur in the presence of EXO1 in S-phase, possibly acting on single-stranded DNA of different resection length. We now include this reference in the discussion to reinforce the idea that the regulation of TMEJ in S-phase and in mitosis are different (Lines 580-586).

2. The study starts from assessing the role of polQ in CSR, but it diverts on the handling persistent DSBs in S phase by TMEJ. Can the authors reconcile this with more emphasis on the importance of these observation in physiological settings ?

We agree with referee #1's comment. Two important findings emerge from our study.

1) The role of Pol θ in CSR, specifically in NHEJ-deficient settings, is here addressed for the first time. This leads to the discovery that Pol θ is dispensable for productive CSR, even though it leads to the formation of unproductive recombination products, thus maintaining overall *Igh* locus stability.

2) We reveal that, in the absence of NHEJ, persistent G1-DSBs are repaired in S-phase by Pol θ in a RHINO- and PLK1-independent manner, distinguishing S-phase TMEJ from mitotic TMEJ.

We have modified both the title and the abstract of our manuscript to better emphasize the physiological function of Pol θ in CSR. We have also modified the discussion and notably added a final paragraph that recapitulates both findings.

Reviewer #2 (Remarks to the Author):

This study by Marton et al. investigates the mechanisms of double-strand break (DSB) repair during class switch recombination (CSR), a process essential for antibody diversification. To this end, they examine CSR activity in various genetic contexts, specifically in the absence of classical non-homologous end joining (NHEJ) factors (XRCC4 and SHLD1) or the alternative/microhomology-mediated end joining (Alt-EJ) factor, Pol theta (Pol θ), both pathways known to contribute to end-joining during CSR. The study particularly focuses on the role of Alt-EJ, mediated by Pol θ , when NHEJ is impaired. To explore this, they analyze the repair of DSBs induced in the G1 phase, where DSBs are typically resolved by either NHEJ or Alt-EJ pathways. The authors show that, in the absence of NHEJ, Pol theta repairs G1-programmed DSBs that have undergone resection. They further demonstrate that this repair occurs during the S/G2 phase of the cell cycle, rather than during mitosis, as it is entirely independent of Rhino, a recently identified mitotic-specific Pol θ interactor.

The study is well conducted, and the results are clearly presented. The data have the potential to advance the field.

I have only few comments:

The conclusion that Pol θ -mediated repair occurs rapidly upon G1 exit is based on mRNA expression and cell proliferation data, and, in my view, represents an overstatement.

We thank referee #2 for his/her comment and now provide more direct evidence that Pol θ -mediated repair of persistent G1-DSBs occurs upon cell cycle entry into S-phase (See response to referee #1 and new Supplementary Figure 10)

Additionally, there is a potential bias in the interpretation, as XRCC4^{-/-}; POLQ^{-/-} cells do not appear to effectively progress through the cell cycle following the induction of RAG-induced DSBs (Fig. 6D). In this context, it might be valuable to use a POLQ inhibitor instead, as this could reduce repair activity while potentially allowing cells to continue progressing through the cell cycle.

Indeed, *p53*^{-/-} *Xrcc4*^{-/-} *Polq*^{-/-} pro-B cells do not effectively progress throughout the cell cycle following the induction of RAG-induced DSBs in G0/G1. This is due to an incapacity to repair DSBs upon cell cycle progression, leading to an accumulation of numerous chromosomes breaks that cause synthetic lethality. This is the core of our previous work published in 2020 (Yu et al, Nat Comm, 2020 Oct 16, PMID: 33067475, DOI: 10.1038/s41467-020-19060-w) and consistent with observations from colleagues the same year in the context of NHEJ/Pol θ - inhibited cells exposed to ionizing radiations (Kumar RJ et al, NAR Cancer, Dec 2020, PMID: 33385162, DOI: 10.1093/narcan/zcaa038).

In the present study, we show that this synthetic lethality is independent of RHINO (*i.e.* *p53*^{-/-} *Xrcc4*^{-/-} *Rhno1*^{-/-} pro-B cells effectively progress through the cell cycle after being exposed to RAG-DSBs in G0/G1 and do not exhibit such massive chromosomal instability), consistent with our conclusion that TMEJ of persistent G1-DSBs is mechanistically distinct from TMEJ in mitosis (also refer to our responses below).

Nevertheless, we tested the ART558 POL θ inhibitor (Pol θ i) on *p53*^{-/-} *Xrcc4*^{-/-} pro-B cell lines. In G0/G1-block and release experiments, Pol θ i treatment recapitulates the dependence on POL θ for repair and survival: no *Igk* V₁₀₋₉₅ J₄ recombination events (Supplementary Fig. 9c) or *Igk*-mediated translocations (Fig. 6h) were detected upon release of *p53*^{-/-} *Xrcc4*^{-/-} pro-B cells treated with Pol θ i, indicating an absence of repair. Treated cells also showed delayed G0/G1 exit after ABLki washoff and impaired proliferation, as assessed by cell cycle FACS analysis and cell count kinetics (Supplementary Fig. 9e-f). These results support the role of POL θ in repair of persistent G1-DSBs and its importance for the survival of NHEJ-deficient cells exposed to G1-DSBs.

We additionally tested POL θ i in the CSR setting in CH12 B cells. Pol θ i treatment slightly impaired overall CSR independently of NHEJ status in CH12 cells. Nevertheless, remarkably, ART558 treatment of *Xrcc4*^{-/-} CH12 B cells phenocopied *Polq*^{-/-} *Xrcc4*^{-/-} CH12 B cells, with resected recombination products no longer detectable in these cells and a concomitant increase in the level of *Igh* instability (Supplementary Fig. 6).

Reading the paper, one might get the impression that the authors claim Pol θ functions during

mitosis is exclusively a Rhino-dependent. This point should be clarified, as it does not accurately reflect the state of the field; the involvement of Rhino remains preliminary as it is currently supported only by a single publication. Using a PLK1 inhibitor could strengthen the evidence for a mitosis-independent role of Pol θ and help consolidate the findings presented in the study. It would also be valuable to generate Rhino $^{-/-}$; Pol $\theta^{-/-}$ cells to further investigate this relationship.

We thank referee #2 for raising this important point.

We now emphasize the role of PLK1 throughout the manuscript and tested a PLK1 inhibitor (PLK1i, volasertib) in our pro-B cell block/release experiments.

Of note, we found that pro-B cells are very sensitive to PLK1 inhibition regardless of NHEJ status and even in the absence of G0/G1-DSB induction (Supplementary Fig. 9a), demonstrating the inhibitor's potency, but also limiting our readouts to recombination efficiency and *Igk* instability. Using a dose of PLK1 inhibitor where >80% of cells survived, we assessed RAG-induced DSB repair in released *p53^{-/-} Xrcc4^{-/-}* pro-B cells. In parallel of PLK1i experiments, we also inhibited POL θ for direct comparison between the two inhibitors (New Supplementary Figure 9 and new Figure 6h). We found that upon PLK1i treatment, contrary to Pol θ i treatment, *Igk* V₁₀₋₉₅ J₄ recombination occurs at levels similar to DMSO controls (Supplementary Fig. 9c), indicating that TMEJ of persistent G1-DSBs occurs independently of PLK1. Consistent with this, metaphase spread analysis showed that TMEJ remains active under PLK1 inhibition, with similar levels of chromosomal translocations detected in released *p53^{-/-} Xrcc4^{-/-}* pro-B cells treated with DMSO or PLK1i, in sharp contrast to released *p53^{-/-} Xrcc4^{-/-}* pro-B cells treated with Pol θ i that displayed high levels of only chromosome breaks (Fig. 6h).

Thus, we now conclude that persistent G1-induced DNA breaks are repaired by TMEJ in a manner independent of RHINO and PLK1 activity.

It is difficult to discern the major novelty of the findings, particularly how they differ from the previous study by the same group, which showed that DSBs generated in G1 in NHEJ-deficient cells are repaired during the following S/G2/M phases by Pol theta (Yu et al., Nat Comm, 2020). This distinction should be addressed and discussed in the manuscript's discussion section.

There are two major findings in this study that were not present in our original work from 2020 and not addressed in the current literature.

1) The role of Pol θ in CSR, specifically in NHEJ-deficient settings, is here addressed for the first time. This leads to the discovery that Pol θ is dispensable for productive CSR, even though it leads to the formation of unproductive recombination products, thus maintaining overall *Igh* locus stability. How CSR functions in the absence of NHEJ has been a debate in the community since 2007 (Yan et al, Nature, 2007 Sept 27, PMID: 17713479, DOI: 10.1038/nature06020 and Soulas-Sprauel et al, J Exp Med, 2007 July 9, PMID: 17606631, DOI: 10.1084/jem.20070255). Whether and to which extent Pol θ is implicated in this physiological setting is here addressed, both *in vivo* in mice and in B cell lines.

2) We here reveal that, in the absence of NHEJ, persistent G1-DSBs are repaired in S-phase by Pol θ in a RHINO- and PLK1-independent manner. In our original paper in 2020, we did

not measure TMEJ upon cell cycle entry into S-phase but only at later time points. This is why we could not conclude on whether TMEJ of G1-DSBs occur in S, G2, or M-phases of the cell cycle. Since 2020, two studies identified the mode of action of Pol θ -mediated repair in mitosis, this in the context of persistent replicative breaks (Gelot et al, Nature, 2023 Sept, PMID: 37674080, DOI: 10.1038/s41586-023-06506-6 and Brambati et al, Science, 2023 Aug 11, PMID: 37440612, DOI: 10.1126/science.adh3694). In the revised manuscript, we provide strong evidence that TMEJ of persistent G1-DSBs occurs in S-phase, independently of RHINO and PLK1 activity, and is thus mechanistically distinguishable from TMEJ of replicative breaks in mitosis.

We have revised the manuscript to better emphasize our findings on Pol θ 's role in CSR (also requested by referee #1) by providing a new title and abstract and, by expanding the discussion

Reviewer #3 (Remarks to the Author):

The manuscript entitled titled "Polymerase theta-mediated end-joining repairs persistent G1-induced DNA breaks in S/G2" by T. Marton et al. presents important findings that significantly advance our understanding of DNA repair pathway choice across the cell cycle. The authors demonstrate that B cells deficient in both XRCC4 and Pol θ , or SHLD1 and Pol θ , accumulate unresolved IgH double stranded DNA breaks. Surprisingly, these cells retain efficient class switch recombination (CSR), highlighting the role of polymerase theta-mediated end joining (TMEJ).

This study reveals that in the absence of core non-homologous end joining (NHEJ) components, Pol θ mediates aberrant end joining characterized by extensive resection, microhomology usage, and inversion events, independently of the RHINO complex. These observations challenge the prevailing view that TMEJ is confined to mitotic phases, demonstrating instead that it can function during S/G2 to resolve persistent G1-induced DNA breaks. This finding redefines the temporal flexibility of TMEJ and expands our understanding of backup repair pathways. As usual from the Deriano lab, the experimental data has been performed rigorously, presented clearly and including all the appropriate controls and corresponding statistical analysis. Nevertheless, a few items need to be addressed before the manuscript is ready for publication.

Specific comments:

1. Clarification needed regarding switching efficiency: The following statement should be revised for accuracy: 'Furthermore, Pol θ 's dispensability extended beyond IgM-to-IgG1 switching, as cells efficiently recombined toward IgG2b, IgG3, and IgA isotypes in both XRCC4-proficient and XRCC4-deficient B cells (Fig. 1E-G)'. In fact, XRCC4-deficient B cells did not switch efficiently to IgG2b, IgG3, and IgA isotypes compared to XRCC4-proficient (WT) B cells. Please rephrase this sentence to reflect the comparative decrease in switching efficiency in the XRCC4-deficient background.

We thank the reviewer for drawing our attention to this sentence.

We have now corrected it to more accurately convey that XRCC4 deficiency leads to a diminished CSR rate. The revised text in lines 175-178 now reads:

Furthermore, Pol θ 's dispensability extended beyond IgM-to-IgG1 switching, as cells efficiently recombined toward IgG2b, IgG3, and IgA isotypes in XRCC4-proficient (WT) (Fig. 1e-g). Although XRCC4-deficient B cells switch less efficiently towards IgG2b, IgG3, and IgA (compared to WT), Pol θ is also dispensable in this genetic setting (Fig. 1e.g).

2. Unclear Smears in Figure 3B: The authors refer to smears below the S–S joints in XRCC4-deficient cells (Figure 3B). However, these smears are not visible in the gel as presented. In contrast, smears are clearly visible in the SHLD1-deficient condition. Please consider repeating the experiment or providing a higher-quality gel image to confirm the results.

We agree with reviewer #3 that LR-PCR smears in *Xrcc4*^{-/-} cells are difficult to visualize, as they are much smaller than in SHLD1-deficient cells. These differences are molecularly relevant: SHLD1 limits resection in both NHEJ-proficient and NHEJ-deficient cells (see Vincendeau et al., Nat Commun, 2022 Jun 28, PMID: 35764636, DOI: 10.1038/s41467-022-31287-3). Hence, in *Xrcc4*^{-/-} cells, the 53BP1/SHLD complex remains active and protects DNA ends from extensive resection, resulting in resected junctional amplicons that are less visible on agarose gels.

Importantly, the gel profiles shown correspond to samples used for PacBio sequencing and read-length data, which confirm the presence of the faint smears in *Xrcc4*^{-/-} activated B cells in the form of resected joints (Supplementary Fig. 3). Moreover, an independent high-throughput sequencing approach (CSR-HTGTS) recapitulates the increased number of resected joints. Therefore, we are confident in our results.

For sake of clarity and avoid confusion for the readers, we have moved the *Xrcc4*^{-/-} agarose gel to the Supplementary Fig. 3a, leaving the SHLD1-deficient profile in the main figure for LR-PCR gel visualization.

3. In Figure 3C, an asterisk appears in the figure without explanation. Please add a corresponding note in the figure legend regarding the asterisk.

We thank reviewer #3 for this comment. We have now added a corresponding note in the figure legend to clarify the meaning of the asterisk in Fig. 3c. Line 1075 now reads: Asterisk (*) represents an unspecific band. We also added this to Supplementary Fig. 3a, containing LR-PCR profiles from all PacBio sequenced samples (Line 1201).

4. Missing Panels in Figure S3: The authors reference panels C and D of Supplementary Figure S3 in the Results section, but these panels are absent from the current version of the figure. Please verify and include the missing panels along with an updated legend.

We thank reviewer #3 for pointing this out. Upon careful verification, the references to panels “c” and “d” in Supplementary Fig. 3 were incorrect. In the “TMEJ generates resected and inversional unproductive joints during CSR” section, we should have referred to panel “a” for LR-PCR replicates (smears) and panel “b” for read length distributions (violin plots). Accordingly, all figure references have been corrected: Supplementary Fig. 3c is now Fig. Supplementary 3a, and Supplementary Fig. 3d is now Supplementary Fig. 3b.

5. The manuscript states that XRCC4 or SHLD1 deletion significantly increases S μ –S μ joints (Fig. 4B, Fig. S4A, B). However, increased S μ –S μ joining is observed only in XRCC4-deficient cells (Fig. S4A). SHLD1-deficient cells show a reduction compared to WT (Fig.

S4A). Please revise this section to accurately describe the data.

We have revised the text to accurately reflect the data. Lines 307-309 now read: In *Xrcc4*^{-/-} we recapitulate the increasing S μ -S μ joints^{23,29,38} while no significant difference is observed upon SHLD1-depletion as compared to WT.

6. To establish that TMEJ is responsible for repairing RAG-induced double-strand breaks (DSBs) specifically in S/G2-phase cells, it is essential to show that TMEJ-type junctions are more prevalent in S/G2-phase cells. This can be achieved by combining cell cycle analysis with molecular characterization of DNA repair junctions. Additionally, it is important to demonstrate that features of Pol θ -mediated repair, such as microhomology, resection, and insertions, are predominantly found in S/G2-phase cells. A cell cycle analysis using FACS sorting to isolate S/G2-phase cells, followed by LR-PCR to amplify and sequence RAG-induced junctions could be performed. Compare the frequency of TMEJ features (microhomology, resection, and insertions) in S/G2-phase cells versus mitotic cells to demonstrate preferential repair by Pol θ in S/G2.

We thank the reviewer for this comment and suggestions, which helped us address concerns raised by the other two referees and now enable us to significantly strengthen our conclusions on the cell cycle analysis of TMEJ.

We have conducted additional experiments that now allow us to demonstrate TMEJ activity in G1/S-phase of the cell cycle (see also response to referee #1). As suggested, we used *p53*^{-/-} *Xrcc4*^{-/-} pro-B cell clones to perform block/release experiments, followed by FACS sorting of G1, S, and G2/M populations, this to assess the timing of TMEJ repair after G1-DSB induction (Supplementary Fig. 10). Cells were collected at day 2 and day 3 post-ABLki washoff, corresponding to the first wave of cells exiting G1 into S-phase (Fig. 6c), and cell fractions were marked using EdU/PI before sorting and analysis. To track TMEJ repair, we assessed *Igk* V₁₀₋₉₅ J₄ recombination and detected joins predominantly in S-phase, with a few arising as early as G1 (Supplementary Fig. 10b). We cloned and sequenced these junctions, revealing TMEJ signatures, including resection and microhomology usage (Supplementary Fig. 10c-d). Together, these results strongly strengthen our conclusion that TMEJ of persistent G1-DSBs operates during the G1/S transition and is not delayed until mitosis.

7. Genotyping Band Size Error in Supplementary Figure S1A; the size of the WT band for XRCC4 genotyping is incorrectly indicated. Please correct the band size.

We thank the reviewer for noting this. Upon verification, the XRCC4 genotyping band sizes indicated on the left side of gel images are now correct and reflex expected band size and not ladder. The Supplementary Fig. 1a has been updated to clearly indicate the correct sizes.

8. Mismatch Between Figure and Text (Figure 2B): In the results part, specifically the paragraph about TMEJ joints a subset of AID-induced DNA breaks during CSR, the authors mention in the text the mean numbers of aberrant metaphases and refer to Figure 2B and supplementary Table S1. However, the authors represented the number of translocations and breaks in Figure 2B. I suggest adding legend for breaks and translocations bars on the figure itself as Figure 5C and refer only to table S1 in the text for the number of aberrant metaphases.

We have revised Fig. 2b by changing the bar colors to white and red and adding a legend to clearly indicate breaks versus translocations. In the text, we now refer to Supplementary Table S1 when reporting the absolute numbers of aberrant metaphases, while Fig. 2b displays the percentages. These changes ensure consistency between figures and the text.

9. Inconsistent Use of Units (bp vs. Kb): The authors use both “bp” and “Kb” when referring to fragment sizes in the same section. For clarity and consistency, please standardize the unit usage throughout the manuscript.

We have standardized the use of kilobases (kb) throughout the manuscript for consistency, particularly in the “TMEJ generates resected and inversional unproductive joints during CSR” section, and all relevant text has been updated accordingly.

10. The authors refer to Figure 4B and 4D when describing the percentage of resected joints in WT and Polq^{-/-} IgG1-stimulated cells. However, this data is shown in Figure 4C and 4D. Please correct the citation accordingly.

Indeed, the percentages of resected joints are shown in Fig. 4b (zoom-in of the Cyl region) and Fig. 4c (histogram of resected ends), whereas Fig. 4d corresponds to inversion/deletion ratios and is not related to resection. We have corrected the text accordingly (Lines 303, 316, 321).

11. The authors described the % of S_μ-S_{γ1} direct and possess MH>4bp joints by citing only Figure 4E. Please also reference Figure 4F, which presents the data for S_μ-S_{γ1} joins with MH >4 bp. The sentence, “GLTs and AID expression levels were assessed across CH12F3 cell lines used (Fig. S7),” appears in the CSR assay in CH12F3 cells lines section. It would be more appropriate to include this sentence in the paragraph related to RT-qPCR for AID and GLT expression.

Thank you. We now refer to Fig. 4 panels “f” (histogram) and “g” (MH length landscape) for joins which exhibit direct and MH>4 bp, respectively (Lines 341, 346, 347). Additionally, the mentioned sentence has been moved from the Cell Line sub-section to the RT-qPCR sub-section of Methods.

12. Please correct the genotype annotations in Supplementary Tables S1, S2, S3, S4 and S5 to maintain consistency with Tables S6 and subsequent tables. For example, “polq” should be written as “polq^{-/-}” and etc...

For clarity and consistency, we now always refer to deletion mutants using “-/-” after the gene name and no longer use shortcuts in supplementary tables.

October 2, 2025

Response to the reviewers' comments

REVIEWERS' COMMENTS

Reviewer #1 (Remarks to the Author):

The authors have addressed the comments and the manuscript is now suitable for publication.

Reviewer #2 (Remarks to the Author):

The manuscript significantly improved during the revision process, particularly with the inclusion of novel data and controls. Altogether this manuscript provides strong evidence for a role of POLQ in repairing programmed G1 breaks during the subsequent S phase rather than in mitosis.

I have no further comments and am pleased to strongly recommend the revised manuscript for publication.

Reviewer #3 (Remarks to the Author):

The authors have addressed all my concerns and modified the manuscript accordingly. I believe the manuscript is ready for publication.

Response to the reviewers' comments. We thank the referees for their careful review of the manuscript, their positive comments on our work and their insightful suggestions for improving our article.